# Giant electric field-induced second harmonic generation in polar skyrmions

Sixu Wang[1,7], Wei Li[1,7], Chenguang Deng[1], Zijian Hong ®[2,3] ✉, Han-Bin Gao[4], Xiaolong Li[5], Yueliang Gu[5], Qiang Zheng ®[4] ✉, Yongjun Wu[2], Paul G. Evans ®[6], Jing-Feng Li ®[1], Ce-Wen Nan ®[1] & Qian Li ®[1] ✉

Electric field-induced second harmonic generation allows electrically controlling nonlinear light-matter interactions crucial for emerging integrated photonics applications. Despite its wide presence in materials, the figures-of-merit of electric field-induced second harmonic generation are yet to be elevated to enable novel device functionalities. Here, we show that the polar skyrmions, a topological phase spontaneously formed in $PbTiO_3/SrTiO_3$ ferroelectric superlattices, exhibit a high comprehensive electric field-induced second harmonic generation performance. The second-order nonlinear susceptibility and modulation depth, measured under non-resonant 800 nm excitation, reach ~54.2 pm $V^{-1}$ and ~664% $V^{-1}$, respectively, and high response bandwidth (higher than 10 MHz), wide operating temperature range (up to ~400 K) and good fatigue resistance (>$10^{10}$ cycles) are also demonstrated. Through combined in-situ experiments and phase-field simulations, we establish the microscopic links between the exotic polarization configuration and field-induced transition paths of the skyrmions and their electric field-induced second harmonic generation response. Our study not only presents a highly competitive thin-film material ready for constructing on-chip devices, but opens up new avenues of utilizing topological polar structures in the fields of photonics and optoelectronics.

Second-harmonic generation (SHG) is a second-order nonlinear optical ($\chi^{(2)}$) process in which two photons with $\omega$ frequency combine into one frequency-doubled ($2\omega$) photon. Since its discovery in 1961, SHG has been widely employed in the realm of lasers and optoelectronic devices as well as in materials and biomedical sciences as the fundamental basis of powerful characterization methods[1–3]. More recently, the advent of integrated photonic platforms has fostered emerging on-chip devices such as all-optical switches, frequency combs, and quantum light sources[4–6]. Emerging SHG devices demand integratable

materials in which nonlinear optical properties can preferably be tuned by an applied electric field to enable electrical modulation and/or reconfiguration functionalities. As mediated by a third-order nonlinear optical ($\chi^{(3)}$) process which imposes no restraints on crystal symmetries, electric field-induced second-harmonic generation (EFISH) has been studied for various materials, including optical crystals[7,8], polymers[9], (strain-free) silicon[10,11], layered metal chalcogenides[12–16] and artificial photonic structures[17,18]. However, these materials in general exhibit weak effective $\chi^{(2)}$ susceptibilities (which is

[1]State Key Laboratory of New Ceramics and Fine Processing, School of Materials Science and Engineering, Tsinghua University, 100084 Beijing, China. [2]School of Materials Science and Engineering, Zhejiang University, 310027 Hangzhou, China. [3]Research Institute of Zhejiang University-Taizhou, 318000 Taizhou, Zhejiang, China. [4]CAS Key Laboratory of Standardization and Measurement for Nanotechnology, CAS Center for Excellence in Nanoscience, National Center for Nanoscience and Technology, 100190 Beijing, China. [5]Shanghai Synchrotron Radiation Facility, Shanghai Advanced Research Institute, Chinese Academy of Sciences, 201204 Shanghai, China. [6]Department of Materials Science and Engineering, University of Wisconsin-Madison, Madison, WI 53706, USA. [7]These authors contributed equally: Sixu Wang, Wei Li. ✉e-mail: hongzijian100@zju.edu.cn; zhengq@nanoctr.cn; qianli_mse@tsinghua.edu.cn

the product of the $\chi^{(3)}$ susceptibility and electric field) and low modulation depths (mostly a few percent per volt), limiting their applications. Alternatively, tunable SHG can be realized in polar, $\chi^{(2)}$-active materials by altering their crystallographic structures or domain microstructures. The switching of ferroelectric domains is accompanied by modulation of SHG responses due to an interference effect between the opposite domains, which can be appreciable depending on specific domain configurations within the illuminated material volumes[19,20]. Large SHG modulation can also be generated through an electrically induced transition between polar and nonpolar phases, as exemplified by BiFeO₃/TbScO₃ superlattices[21] and MoTe₂ monolayers[15]. However, such processes typically occur at mesoscopic scales and are determined by the phase front or domain wall propagation kinetics, invariably resulting in hysteretic and sluggish modulation responses further to be plagued by fatigue phenomena. The combination of hysteresis and slow modulation significantly reduces the utility of the high SHG modulation depths exhibited by conventional polar materials.

Recent observations of topological polar structures bring new opportunities for discovering and designing optoelectronic functionalities. These polar structures include, for instance, vortices, skyrmions and merons hosted in PbTiO₃/SrTiO₃ superlattices or multilayers[22–29]. The polarization configurations commonly feature a nanosized periodic unit with swirling electric dipoles and long-range in-plane ordering of the polar units. Local regions with highly frustrated polarization are stabilized in these polar structures, resembling Néel or Bloch types of domain walls. Emergent properties result from the rich structural characteristics, including negative permittivity[30], SHG circular dichroism[31] and sub-terahertz collective dynamics[32]. Several studies have also examined the in situ polarization evolution of polar vortices and skyrmions, indicating their responsiveness to external stimuli with large alterations of the polarization magnitude[33–36]. These results thus hint at the potential of topological polar structures for SHG modulation.

Here, we introduce a polar-skyrmionic EFISH effect, shown schematically in Fig. 1a. In the ground state, the skyrmions collectively are $\chi^{(2)}$-inactive due to the pseudo-centrosymmetry exhibited by the dipoles of each skyrmion and the interfacial c-domain region surrounding it. Symmetry breaking results from applying an electric bias to tip the balance of the dipoles and engenders a strong SHG response. This process does not involve the nucleation or motion of domain walls and thus can be tuned rapidly and reversibly with electric bias,

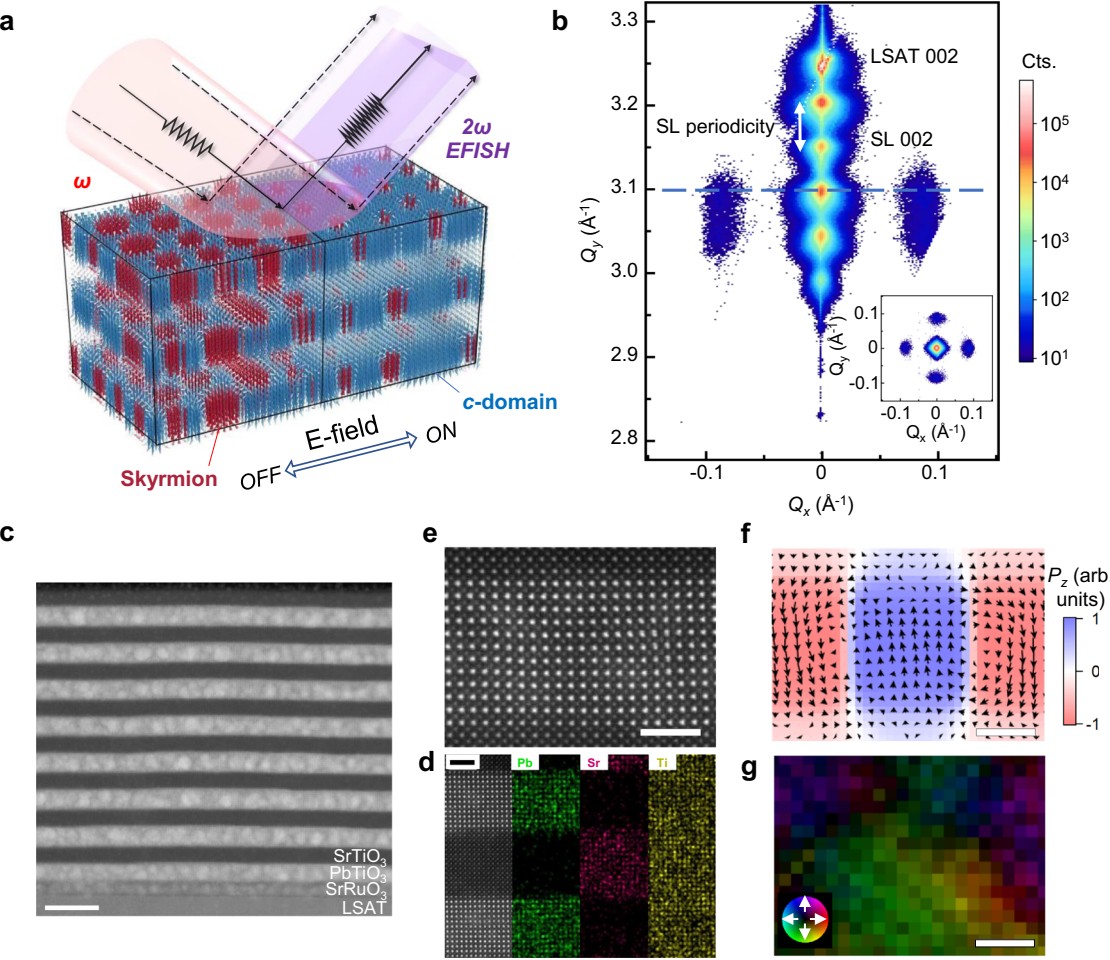

**Fig. 1 | Structural characterizations of polar skyrmion-hosting PbTiO₃/SrTiO₃ superlattices. a** Phase-field model-adapted schematic of an electric field-induced transition (1 MV cm⁻¹ for the right part) in the polarization vector (arrows) configuration of polar skyrmions that engenders an enhanced second-harmonic generation response. **b** *H0L*-slice (at $Q_y = 0$ Å⁻¹) X-ray reciprocal space map around the LSAT 002 reflection for a [(PbTiO₃)₁₄/(SrTiO₃)₁₆]₈ superlattice, with the arrow denoting the superlattice (SL) periodicity along the $Q_z$ direction. Inset: *HK0*-slice (at $Q_z = 3.097$ Å⁻¹) reciprocal space map showing the in-plane skyrmion periodicity with four strong lobes along the *H00/0K0* directions. **c** Cross-sectional HAADF-STEM image of [(PbTiO₃)₁₄/(SrTiO₃)₁₆]₈ superlattices, displaying distinct intensity contrast for the constituent layers as marked. Scale bar = 20 nm. **d**, Energy-dispersive X-ray spectroscopy mapping for Pb, Sr and Ti. **e–g** High-resolution HAADF image (**e**) and corresponding polar displacement (**f**) and electric field (**g**) distribution maps for a PbTiO₃ layer region containing a skyrmion. The phase-field simulated $P_z$ distribution is overlaid in (**f**). **g** is extracted from the measured DPC images. Scale bar = 2 nm.

yielding giant SHG modulation depths that are not available in spatially simpler polarization configurations. Other advantages of the exotic polarization configuration of the skyrmions, described below, also contribute to a highly appealing overall EFISH performance. PbTiO$_3$/SrTiO$_3$ superlattices (and similar ferroelectric/dielectric systems of other materials) can be readily adopted for constructing on-chip optoelectronic devices, either by direct epitaxial growth or through the transfer and integration of released oxide membranes. Our study advances the understanding of nonlinear optical properties of topological polar structures and may lead to new directions in integrated photonics.

## Results

### Superlattice growth and characterizations

As a model system, we focus on ~96 nm $[(PbTiO_3)_{14}/(SrTiO_3)_{16}]_8$ superlattices epitaxially grown on (001)-cut $(LaAlO_3)_{0.3}$-$(SrAl_{0.5}Ta_{0.5}O_3)_{0.7}$ (LSAT) substrates buffered with a conducting 5 nm SrRuO$_3$ layer. In comparison with the polar skyrmion-hosting superlattices previously realized on SrTiO$_3$ substrates, we select LSAT substrate as it exhibits much lower absorptivity at terahertz frequencies and larger contrast in the refractive indices with PbTiO$_3$ and SrTiO$_3$[37,38], facilitating the guided propagation of light in the superlattices. We have controlled the growth conditions such that the substrate misfit strain is partially (by about 1%, biaxially) relaxed within the SrRuO$_3$ buffer layers via a low-concentration array of edge dislocations at the SrRuO$_3$/LSAT interface, as confirmed by cross-section scanning transmission electron microscopy (STEM) imaging and X-ray reciprocal space mapping (RSM) about the LSAT 013 and 103 in-plane reflections (Supplementary Fig. S1). Relaxing of the epitaxial mismatch within the SrRuO$_3$ layer is critical to achieving a sufficiently low defect density and strain gradient in the subsequent growth of the superlattice. A three-dimensional (3D) RSM about the LSAT 002 reflection (Fig. 1b), together with the two in-plane RSMs, reveals excellent crystallinity of the resultant superlattices. Well-defined thickness fringes are observed along the out-of-plane direction in agreement with the designed layer periodicity of ~12 nm (that is, 30 unit cells). Along the in-plane directions, diffraction satellites appear at wavevectors separated from the *OOL* rod by ~0.083 Å$^{-1}$, which equates to a real-space periodicity of ~7.6 nm, with a fourfold lobe pattern of the satellites. These features are apparent in the *HKO* RSM (Fig. 1b, inset). The diffraction features indicate the occurrence of well-populated polar skyrmions with a large extent of in-plane ordering along the [100] and [010] directions.

The real-space configuration of the skyrmions was probed using cross-section STEM acquired in both high-angle annular dark-field (HAADF) and differential phase contrast (DPC) modes, as presented in Fig. 1c–g. Large-scale HAADF images (Fig. 1c) show uniform layers of PbTiO$_3$ and SrTiO$_3$ with clear interfaces. Elemental mapping results confirm no interdiffusion between the two constituent layers (Fig. 1d), again attesting to the high growth quality. The HAADF intensity contrast reflects the locations of cation columns, from which polar displacement vectors can be derived. Local electric field information can be semi-quantitatively extracted from the vectorial DPC signals. Correlative distribution maps of local polar displacement and electric field vectors are illustrated for a region of a single PbTiO$_3$ layer within the superlattice in which there is a skyrmion with two interfacial *c*-domains (Fig. 1e–g). Line profiles of the *z* components $P_z$ and $E_z$ are depicted around the center of the skyrmion in Supplementary Fig. S1e. The polarization and electric field alternate with opposite directions across the skyrmion walls. In the imaged region, the polarization vectors form clockwise/anti-clockwise swirling patterns and have greatly suppressed polarization magnitude, reminiscent of Néel-type domain walls. Altogether, these structural characteristics are consistent with the previous accounts of polar skyrmions[25,28,30], suggesting an optimal testbed of our PbTiO$_3$/SrTiO$_3$ superlattices for exploring their nonlinear optical properties.

### EFISH response of the polar skyrmions

The EFISH effect of the skyrmions was measured at a macroscopic device level using thin-film capacitor devices. The test devices consisted of all-oxide, symmetric SrRuO$_3$/superlattice/SrRuO$_3$ capacitors with a diameter of 100 μm fabricated by chemical etching lithography (see details in "Methods"). The top SrRuO$_3$ layer serves both as an electrode and a semi-transparent optical window for SHG measurements. The normal incidence geometry allows in-focus imaging via laser scanning[39], as shown in Supplementary Fig. S2 and the SHG intensity map inset of Fig. 2a. The EFISH response of the capacitors has a distinct and uniform (except for the electrode boundaries) SHG intensity contrast for the biased region and a constant background signal for the outside region. Focusing at a single spot on the capacitor, the SHG intensity as a function of bias voltage and angles of the input/output light polarization (that is, SHG polarimetry) can be acquired, as shown in Supplementary Fig. S2. These measurements yield systematic evolution trends that are well reproducible among a few tens of capacitors tested in this study, signifying the robustness of the EFISH

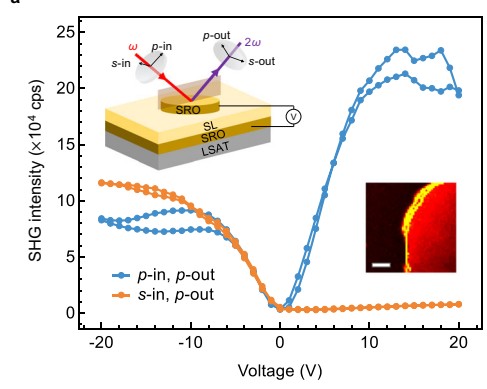

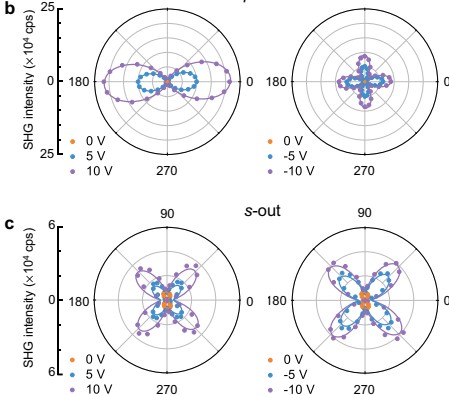

**Fig. 2 | EFISH response of the polar skyrmions measured using capacitor devices. a** Measured SHG intensity as a function of applied voltage for different polarization states (*p*-, parallel and *s*-, perpendicular to the plane of incidence) of the input fundamental light. The bias voltage sequence in each cycle was from 0 to 20 V, then to −20 V and back to 0 V. Insets: schematic of the 45° measurement geometry of in situ SHG on a SrRuO$_3$/superlattice/SrRuO$_3$ (SRO/SL/SRO) capacitor

device, and a SHG image near the electrode boundary measured in the normal incidence geometry. Scale bar = 10 μm. **b, c** Polar plots of the input polarization angle-dependent SHG intensity measured under (**b**) 0° (*p*-out) and (**c**) 90° (*s*-out) output polarization conditions. Lines are model-fitting results to the measured data points. An average laser power of ~200 mW was delivered onto the samples in these measurements.

effect. Nevertheless, we have determined that the polar skyrmions can be more effectively probed in the 45° incidence geometry (see Fig. 2a, inset), as opposed to the normal incidence one, because their dominant, out-of-plane polarization vectors can couple with a non-zero $z$-component of the light electric field in the former case, and thus shifted the emphasis to addressing the former results.

Figure 2a presents typical bias voltage-induced evolution trends of the SHG response. Under the $p$-in/$p$-out (light polarization) conditions, the SHG intensity in Fig. 2a increases from the background level (that is, weak to no definite response at 0 V) rapidly until an inflection point at -10 V, after which the increasing trend slows down with gradual saturation and mild signs of decline set in at -15 V. For negative biases (from 0 to −20 V), a similar trend is observed albeit with a lower enhancement factor (-0.3 times) of the maximum SHG intensity. Such evolution trends are found to be fully reversible and almost hysteresis-free during each cycle (especially within the ±10 V range); applying larger voltages could quickly result in irreversible degradations of the capacitors due to elevated leakage currents.

The results in Fig. 2 highlight that the PbTiO₃/SrTiO₃ superlattices with polar skyrmions have a giant EFISH modulation depth. The modulation depth, defined as $\Delta I_{2\omega}(V_{bias})/I_{2\omega}(V_{bias}=0)$, reaches a peak value of -664% V$^{-1}$ at 7 V. Such a giant modulation effect is entirely absent in single-layer PbTiO₃ ($c$-domain structure) and SrTiO₃ epitaxial films measured using the same methods (Supplementary Fig. S3) and thus is a result of the polarization configuration of the PbTiO₃/SrTiO₃ superlattices rather than a consequence of the individual properties of the component layers.

The polar-skyrmionic EFISH effect also exhibits rich anisotropic behaviors, as is evident in the dramatic difference between the $p$-in and $s$-in conditions at positive biases in Fig. 2a. Systematic insight into the anisotropy was obtained from the SHG polarimetry analysis. Figure 2b, c presents input polarization-dependent polar plots measured under $p$-out and $s$-out conditions, respectively. The $p$-out results (Fig. 2b) have distinct pattern symmetries for positive (fourfold) and negative (twofold with missing 90°-oriented lobes) voltages. The magnitude of the $p$-out response is also different between positive and negative voltages, with a larger magnitude at positive bias. The $s$-out results (Fig. 2c) have similar fourfold patterns with 45°-oriented lobes for both polarities. For each bias polarity, these pattern symmetries appear to be invariant with the magnitude of the voltage, and thus can be fitted using a second-order $\chi^{(2)}$ susceptibility tensor with fixed ratios between the coefficients (Supplementary Note 1). Through simultaneous fitting of the $p$-out and $s$-out polarimetry patterns, we have established that the $\chi^{(2)}$ tensors for the positive and negative polarities are:

$$\chi^{(2)}(V>0) = \begin{pmatrix} 0 & 0 & 0 & 0 & 0.57 & 0 \\ 0 & 0 & 0 & 0.57 & 0 & 0 \\ 0.04 & 0.04 & -2.05 & 0 & 0 & 0 \end{pmatrix} \text{arb. units and}$$

$$\chi^{(2)}(V<0) = \begin{pmatrix} 0 & 0 & 0 & 0 & 0.62 & 0 \\ 0 & 0 & 0 & 0.62 & 0 & 0 \\ 1.00 & 1.00 & -0.24 & 0 & 0 & 0 \end{pmatrix} \text{arb. units,}$$

respectively. The ratios between the $\chi^{(2)}_{31}$, $\chi^{(2)}_{33}$, and $\chi^{(2)}_{15}$ for the negative polarity are very close to those of PbTiO₃ bulk crystals (4$mm$ point group)[40,41]. The positive polarity, however, is associated with a markedly enhanced $\chi^{(2)}_{33}$ and diminished $\chi^{(2)}_{31}$. This suggests that the emergence of SHG contributors in the latter is more complex than a single $c$-domain scenario. Furthermore, we have quantified these nonlinear optical coefficients using a reference LiNbO₃ sample (see Supplementary Fig. S4 and Supplementary Note 2). The largest induced $\chi^{(2)}_{33}$ in the PbTiO₃ layers and the global SHG efficiency (occurring at -14 V) are about 54.2 pm V$^{-1}$ and $9.3 \times 10^{-11}$ W$^{-1}$, respectively. This $\chi^{(2)}_{33}$ is larger

than the bulk value (-17 pm V$^{-1}$) and primarily accounts for the giant modulation depth under the $p$-in/$p$-out conditions.

## Structural mechanisms of the EFISH effect

Synchrotron-based in situ RSM measurements, in conjunction with phase-field simulations, reveal the microscopic processes underlying the EFISH modulation. The fact that we used a large X-ray beam footprint (see "Methods") causes diffraction signals to include contribution from the unbiased regions and introduces uncertainties in the quantitative analysis of the bias-induced behavior, but offers a macroscopic view of it with good statistics. From the reconstructed 3D-RSMs (Fig. 3a), we extract the profiles of the first-order $OOL$ superlattice diffraction peak and the intensities of the $HOO$ and $OKO$ skyrmion satellites. Figure 3b illustrates the splitting of the superlattice peak at ±15 V due to the occurrence of a new peak at the lower $Q_z$ side, which corresponds to an out-of-plane lattice expansion by -0.25%. This signifies a process of the polarization vectors rotating towards the applied field direction, that is, a transition from the skyrmions to a single $c$-domain state with an upward or downward $P_z$. Accordingly, the satellite intensities decrease gradually with increasing bias and indistinguishably between the $H$- and $K$-directions (Fig. 3c); meanwhile, no shifts of the satellite wavevectors nor new satellites are observed, suggesting that the superlattice preserves the average in-plane ordering of the pristine state with 4$mm$ macroscopic symmetry. These diffraction results qualitatively corroborate the SHG polarimetry analysis but notably, display no obvious difference between negative and positive biases as would be expected from Fig. 2b.

The real-space configuration of the skyrmions, however, takes different transition paths between the two polarities, even if they can ultimately be converted into single $c$-domains at a high electric field along either direction. Figure 3d presents the phase-field results at ±9.4 V for comparison (see also Supplementary Movie S1 and S2); here we show the distributions of the Pontryagin density $q$, the surface integral of which is the topological number (= ±1) of the skyrmion. Upon application of negative bias, the skyrmions expand and coalesce, leading to a continuous $c$-domain matrix randomly embedded with much elongated, stripe-like (and reduced in quantity) skyrmions. By contrast, the configuration of the skyrmions is better protected against positive biases by shrinking on-site individually (thus maintaining their quantity) and more coherently until reaching a high voltage (>14 V). Such difference stems from the upward $P_z$ direction of the skyrmion center in the pristine state, which is due to the epitaxial growth conditions[42]. We further speculate that the initial state can be selected by modifying the growth conditions. This built-in asymmetry of the superlattices determines the asymmetric-in-bias transition paths at the nanoscale. Note that, over larger length scales, the simulation box-averaged values of the three polarization components follow similar evolution trends for the two polarities (Supplementary Fig. S5), consistent with the diffraction results. Note also that, the field induced skyrmion evolution path in this study agrees qualitatively well with several previous studies with indirect measurements and theoretical calculations[30,43], as well as direct in situ TEM observations[36].

Based on the microscopic processes observed using X-ray diffraction and through phase-field simulations, we find that polar skyrmions cause the EFISH effect through a delicate, skyrmion wall-involved mechanism. As depicted in Fig. 3e, the pristine-state configuration exhibits pseudo-centrosymmetry due to a balance between the skyrmions and interfacial regions (near-zero net $P_z$), and thus is SHG inactive, consistent with the experimental results. The applied out-of-plane electric field directly couples with the PbTiO₃ lattice and causes switching of its electric dipoles to form larger portion of $c$-domains. Therefore, the skyrmions constituted by those dipoles expand or shrink according to the field direction, displacing their wall positions at which the inversion centers are located (nonzero net $P_z$). This process breaks the centrosymmetry and activates

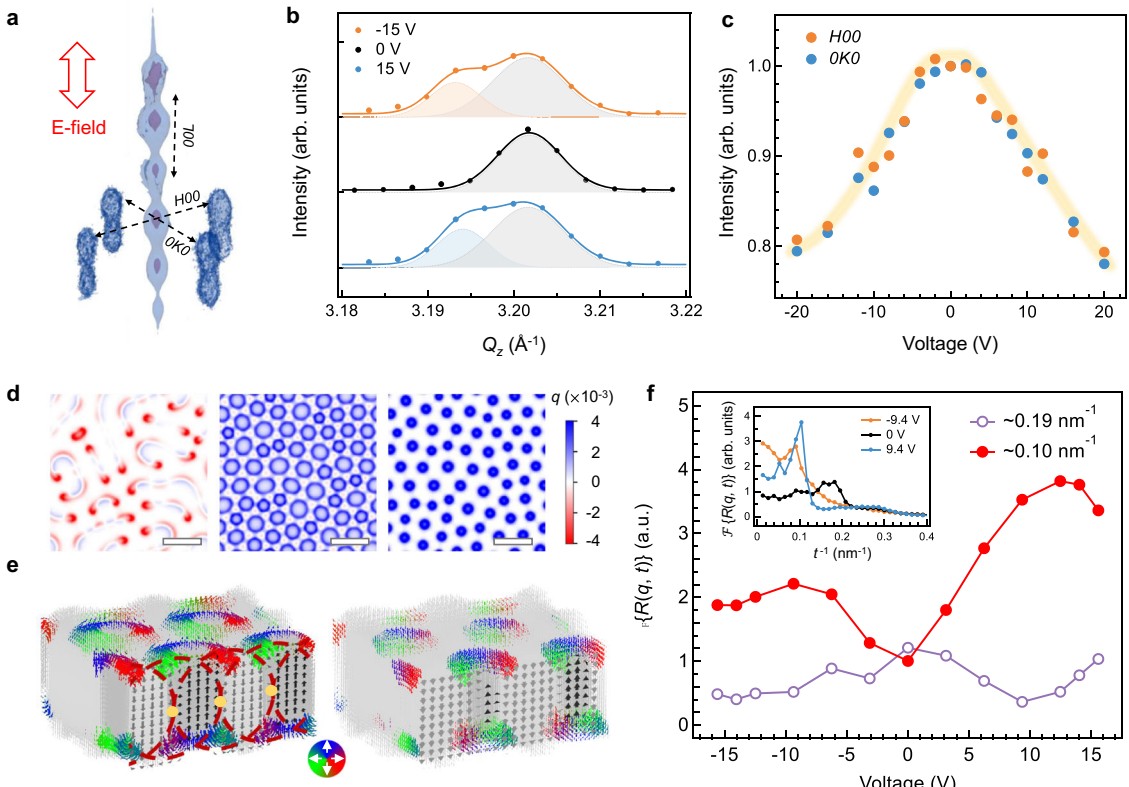

**Fig. 3 | Microscopic mechanisms of the EFISH effect. a** 3D reciprocal space maps near the 002 reflection of the $[(PbTiO_3)_{14}/(SrTiO_3)_{16}]_8$ superlattice, denoting the orientations of applied electric field and crystallographic axes for in situ RSM studies. **b** $00L$ diffraction patterns around the first-order superlattice peak extracted from the 3D-RSMs and corresponding peak-fitting results for different applied voltages. **c** Evolution of the volume-summed diffraction intensities of the skyrmion satellites along the $H00/0K0$ directions as a function of applied voltage. The light yellow line serves as a guide for the eye. **d** Pontryagin density maps for −9.4 V (left), 0 V (middle) and 9.4 V (right) extracted from the phase-field models, highlighting the domain wall regions of the skyrmions. Scale bar = 10 nm. **e** Illustrations for the origin and breaking manner of the pseudo-centrosymmetry in the polar skyrmions under (left) zero and (right) positive biases, respectively. The red circle delineates a centrosymmetric unit, and the yellow dot marks the inversion center. **f** Fourier transform amplitudes of the autocorrelation function of the Pontryagin density, $\mathcal{F}\{R(q, t)\}$, at the -0.103 nm⁻¹ and 0.190 nm⁻¹ spatial frequencies as a function of bias voltage, obtained from the phase-field models. Inset: the spatial frequency spectra of the Fourier amplitudes for selected biases.

the SHG response. To account for the process statistically from the simulated configurations, we calculate the Fourier transform of the autocorrelation function of the Pontryagin densities (Fig. 3f, Supplementary Fig. S6, and Supplementary Note 3). The spatial frequency spectra of the Pontryagin densities exhibit correlation peaks at ~ 0.190 nm⁻¹ and 0.103 nm⁻¹, corresponding to the real-space distance between the two opposite walls of each skyrmion (the intra-skyrmion correlation) and that between the closest pair of skyrmions (the inter-skyrmion correlation), respectively. The intra-skyrmion correlation is suppressed by electric fields, in line with the wall displacement manner of symmetry breaking. The bias-dependence of the inter-skyrmion correlation exhibits a trend highly resembling the observed EFISH trends (cf. Fig. 2a, the $p$-in/$p$-out curve), and thus represents a previously hidden order parameter associated with the EFISH modulation.

To relate the measured $\chi^{(2)}$ to the above structural mechanisms, we have developed a minimal model containing two separate contributors to the SHG process: interfacial $c$-domains and skyrmion walls. At negative biases, due to the weak correlation and the reduced quantity of the skyrmions, the contribution of the skyrmion walls to the global $\chi^{(2)}$ is incoherent and insignificant compared to the $c$-domains, as is indeed evidenced by the SHG polarimetry results. By contrast, the inter-skyrmion correlation is strong and the skyrmion walls are ordered at positive biases; therefore, their contribution to the global $\chi^{(2)}$ can coherently build up to compete with the $c$-domains. The $\chi^{(2)}$ tensor for a circular skyrmion can be derived in a closed form

based on that of Néel-type domain walls[44,45], which also conforms to a $4mm$ point-group symmetry with non-vanishing $\chi^{(2)}_{31}$, $\chi^{(2)}_{33}$, and $\chi^{(2)}_{15}$ (Supplementary Note 4). The skyrmion walls thus should have an anomalously large $\chi^{(2)}_{33}$, in addition to a negative $\chi^{(2)}_{31}$ that cancels with that of the $c$-domains. Note that at the atomistic level, the SHG susceptibility is rooted in the anharmonicity of electronic structure, the calculation of which for the present system is beyond our phase-field framework and yet prohibitive using first-principle methods[46]. We postulate that the frustrated polarization at the skyrmion walls modulates the electronic structure with large anharmonicity along the out-of-plane direction. As a counterpart to the skyrmion walls, the polar vortex cores were found to exhibit partial reduction of $Ti^{4+}$ to $Ti^{3+}$ ions[34], and qualitatively dissimilar $\chi^{(2)}$ tensors for the $a_1/a_2$-domain and vortex phases were also confirmed[47]. We further point out that other extrinsic mechanisms, such as light interference and reflection by the skyrmion walls[48,49] and superlattice interfacial effects[50], might also influence the EFISH response here, but given the large difference between the skyrmion size and probing wavelength, an effective approach is currently lacking to substantiate these subwavelength effects.

## EFISH characteristics for device operations
We next address several key characteristics of the polar-skyrmionic EFISH effect closely pertaining to its application in optoelectronic devices. These measurements also provide further microscopic insights. Figure 4a presents the temperature evolution trend of the

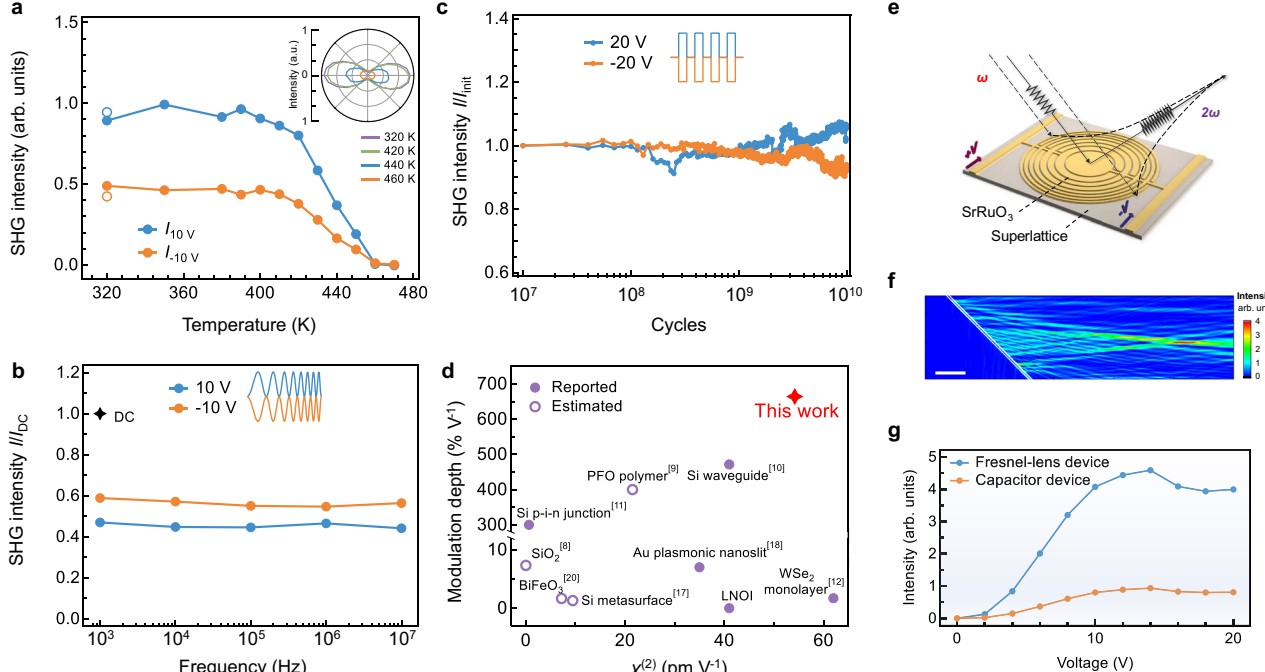

**Fig. 4 | Characteristics of the EFISH effect and an exemplar device design.**
**a** Temperature dependent SHG intensities measured at applied voltages of 10/
−10 V under the *p*-in/*p*-out conditions. The SHG intensity after cooling from 470 K is
shown as hollow dots. Inset: Polar plots of the input polarization angle-dependent
SHG intensity for selected temperatures, measured at 10 V under the *p*-out condi-
tion. The intensities are normalized to the same value scale. **b** SHG intensity as a
function of a.c. frequency of sinusoidal (0 V to ± 10 V peak-peak) voltage wave-
forms. For both polarities, the SHG intensity is normalized against its d.c. value.
**c** SHG intensity as a function of voltage cycle number measured using 10 MHz, 0 to
± 20 V square voltage waveforms, under the *p*-in/*p*-out polarization conditions.

**d** Comparison of the modulation depth and nonlinear susceptibility $\chi^{(2)}$ of $(PbTiO_3)_{14}$/
$(SrTiO_3)_{16}$ superlattices with other materials from literatures. Part of the data points
were estimated values as marked. **e** Schematic of a Fresnel-lens type device that can
focus EFISH signal. **f** Simulated SHG intensity distribution in the incidence plane of
the Fresnel-lens device (shown as the 45° line on the left) excited by a fundamental
light along the -*y* direction. The output light propagates along the +*x* direction,
showing a concentrated intensity at the designed focal length of ~60 μm. Scale bar
= 10 μm. **g** SHG intensity at the focal point of the Fresnel-lens device as a function of
bias voltage (the same value for both polarity of electrodes), in comparison with the
single capacitor device.

SHG intensities measured at d.c. voltages of ±10 V. For both bias
polarities, the SHG intensity decreases as temperature rises and
approaches to the zero-field level above ~450 K, then followed by a
full recovery after cooling down. This trend indicates a reversible
phase transition behavior and sets an upper temperature limit for
applications[51]. Note that the same polarization dependencies during
heating are observed from the polarimetry results (Fig. 4a, inset) and
suggest that the electrically induced transition paths of the sky-
rmions are maintained. This is corroborated by phase-field simula-
tions which reveal an onset of thermal destabilization of the
skyrmions above ~500 K (Supplementary Fig. S5).

Figure 4b presents the frequency dependence of the a.c. EFISH
response, measured using sinusoidal voltage waveforms within the
$10^3$–$10^7$ Hz range. The effective SHG intensities counted over a fixed
period are close to half the d.c. intensities at ±10 V (as can be derived
by integrations of the results in Fig. 2a) and show a marginal decline up
to $10^7$ Hz. This implies that the underlying microscopic processes of
the skyrmions are faster than a time scale of at least 50 ns. Higher
frequency measurements are not available at present but in light of the
flat trends for the measured range, an operating bandwidth in the
gigahertz scales can arguably be inferred. Furthermore, the cycling
fatigue properties are evaluated at high voltages of ±20 V using 10 MHz
square waveforms (that is, on/off switching), as presented in Fig. 4c.
During a typical duration of $10^{10}$ cycles, the SHG intensity variations are
found to be within the range of ±10% of the initial intensity (even with
the sample drifts uncorrected), indicating a remarkable stability.
Intuitively, both the observed fast and fatigue-resistant EFISH response
of the skyrmions can be associated with the small length scale (few-
unit cell), cooperative manners of ion motion in their polarization

transition processes, akin to the emergence of sub-terahertz collective
vibration modes in polar vortices[32].

The above measurements feature a relatively wide working tem-
perature range, high dynamical response bandwidth and long cycling
life of the polar-skyrmionic EFISH effect, all desirable for device
applications such as high-performance nonlinear electro-optic mod-
ulators. In Fig. 4d, we compare $(PbTiO_3)_{14}$/$(SrTiO_3)_{16}$ superlattices with
multiple other types of materials or devices as regards the nonlinear
susceptibility $\chi^{(2)}$ and EFISH modulation depth. The $PbTiO_3$/$SrTiO_3$
superlattice system exhibits the highest level of modulation depth
combined with a strong $\chi^{(2)}$. The $\chi^{(2)}$ value is especially competitive
among thin-film materials and surpasses the LNOI (LiNbO₃-on-insu-
lator) platform. Note that some of the materials or devices included
here realize large figure of merits by exploiting intrinsic or extrinsic
resonance enhancement mechanisms which however are constrained
to narrow wavelength ranges, for example, the electronic interband
transitions in WSe₂ and the PFO conjugate polymer, and the plasmonic
resonance in Au nanoslit cavities[9,12,18]. As to the skyrmions, the EFISH
response is expected to be broadband from mid-infrared frequencies
(above the phonon bands) to the optical bandgap (~3.3 eV), and those
resonance mechanisms may also be incorporated to augment the
response for a particular wavelength range.

To illustrate the SHG modulator applications, we propose a
Fresnel-lens type device with electrically tunable focusing function.
As shown in Fig. 4e, the device operates in the 45° incidence geo-
metry and consists of 4 pairs of concentric $SrRuO_3$ ring electrodes
with varying widths and alternately biased by positive and negative
voltages. We then predict the device output performance using
the measured bias-dependent $\chi^{(2)}$ as an input for the finite-element

modeling. From the simulated intensity distribution profile (Fig. 4f), the SHG signals form a focal point at a length of ~60 μm due to an interference between the oppositely biased areas of the superlattice. The SHG intensity is enhanced at the focal point by a factor of 4 compared to the single capacitor device (Fig. 4g), and the in-between modulation levels can be reached through different combinations of bias voltages. The focal length and enhancement factor of Fresnel-lens devices can be adjusted by varying the geometric parameters. Furthermore, we envisage that an in-plane waveguide type device can be fabricated on the superlattice films with a periodic electrode pattern for biasing alternately or a metasurface structure, thereby potentially achieving high conversion efficiencies through quasi-phase matching[10,52].

## Discussion

Here, we point up that the observed EFISH phenomena reflect the topological protection attribute of skyrmions in ferroelectric/dielectric superlattices. Under applied electric fields, the skyrmions deform and/or shift continuously, yet remain topologically equivalent, until the input electrostatic energy surpasses an energy barrier to break the topological protection. This embodies a general strategy to manipulate the nonlinear optical properties of topological polar structures. Furthermore, $PbTiO_3/SrTiO_3$ superlattices can be re-engineered to tune the quasiparticle interaction energies and ordering states of the skyrmions through the misfit strain and layer makeup[53–55]. For instance, using a more tensile substrate of $SrTiO_3$ may stiffen the skyrmionic interaction and decline the EFISH modulation slope; reducing the $SrTiO_3$ component layer thickness may directly boost the global efficiency due to increased $PbTiO_3$ volume fraction. Other topological phases such as polar vortices, dipole waves and other structures should also be explored for electrically tunable SHG properties, though the likely phase coexistence (e.g., with $a_1/a_2$-domains on $DyScO_3$ substrates) can play an adverse role by involving the hysteretic phase transition processes[33]. More radically, new superlattice or multilayer ferroelectric systems can be designed to stabilize polar skyrmions with engineered electronic anharmonicity (e.g., through ion doping), fusing an intrinsically enhanced $\chi^{(2)}$ with the agile tunability of the skyrmionic configurations.

In conclusion, we have demonstrated the electric field induced second-harmonic generation of the polar skyrmions based on $(PbTiO_3)_{14}/(SrTiO_3)_{16}$ superlattices, and elucidated the microscopic mechanisms through a combination of in situ experiments and phase-field simulations. The fabricated capacitor EFISH devices yield a remarkable comprehensive performance: a modulation depth of ~664% $V^{-1}$, $\chi^{(2)}$ of ~54.2 pm $V^{-1}$ and SHG efficiency of ~9.3 × $10^{-11}$ $W^{-1}$ under 800-nm excitation, reinforced by the other merits such as large response bandwidth, good cycling fatigue resistance and wide operating temperature range. The form of thin-film superlattices presents multiple degrees of freedom for further tuning of the EFISH properties, in addition to an integratable material. All these place the polar skyrmions a technologically competitive system, and may revive the long-lasting interest of ferroelectrics in the fields of integrated photonics, metamaterials and optoelectronics.

## Methods
### Sample fabrication
$[(PbTiO_3)_{14}/(SrTiO_3)_{16}]_8$ superlattices were grown on (001)-cut $(LaAlO_3)_{0.3}$-$(SrAl_{0.5}Ta_{0.5}O_3)_{0.7}$ (LSAT) single crystal substrates by pulsed laser deposition (PLD), with a 5-nm buffer layer of $SrRuO_3$ serving as the bottom electrode. A 248 nm KrF excimer laser (COMPex205, Coherent) was used to ablate ceramic targets of $PbTiO_3$, $SrTiO_3$, and $SrRuO_3$ with the laser fluence and repetition rate as: 1.3 J cm$^{-2}$/5 Hz, 1.7 J cm$^{-2}$/3 Hz and 1.5 J cm$^{-2}$/10 Hz, respectively. The $SrRuO_3$ and superlattice layers were grown in an oxygen pressure of 25 Pa and 10 Pa, respectively, and

the growth temperature for all layers was 650 °C. After deposition, the superlattice films were annealed for 10 min at 650 °C followed by cooling to room temperature under an oxygen pressure of 20 kPa. To fabricate capacitor test structures, a $SrRuO_3$ capping layer of ~20-nm thickness was deposited with the same conditions as the bottom $SrRuO_3$ layer onto the superlattice films. These $SrRuO_3$-capped films were then subjected to ultraviolet optical lithography with the extra $SrRuO_3$ layer removed by dissolving in ~15 g L$^{-1}$ $NaIO_4$ water solution.

### Scanning transmission electron microscopy (STEM)
Cross-sectional thin specimens of the superlattice were prepared using a focused ion beam on a dual-beam microscope (Scios2, Thermo Fisher Scientific) or ion milling (PIPS II, Model 695, Gatan) after mechanical polishing. STEM observations were performed on a double-aberration-corrected transmission electron microscope (Spectra 300, Thermo Fisher Scientific), equipped with a field-emission electron source and operated at an accelerating voltage of 300 kV. Energy-dispersive X-ray spectroscopy (EDX) data was collected using a Super-X detector equipped on the microscope. High-angle annular dark-field (HAADF)-STEM and differential phase contrast (DPC) images were collected with a probe convergence angle of 26.0 mrad. HAADF images were collected with an inner collection angle of 31 mrad, and DPC imaging was performed using a segmented detector with a collection angle of 7–29 mrad. The local electrical field distribution was approximately evaluated from the DPC data, based on quantification of electron beam deflection[56,57]. Positions for all atomic columns in the HAADF images were obtained by 2D Gaussian fits using Matlab codes[58]. Displacement vectors of B-site columns ($D_B$) were calculated based on local offsets of their sublattice relative to the A-site sublattice. The direction of spontaneous polarization ($P_s$) in each unit cell is opposite to the direction of B-site displacement. Values of $P_s$ can be evaluated based on an empirical linear relation between $P_s$ and B-site vector $D_B$, i.e. $P_s = \kappa^* D_B$, where $\kappa$ is a constant of 2.50 μC cm$^{-2}$ pm$^{-1}$ for the $PbTiO_3$ systems[59].

### Synchrotron X-ray diffraction (XRD)
Three-dimensional reciprocal space mapping (3D RSM) and in situ electric field dependent RSM were performed at the BL02U2 beamline of the Shanghai Synchrotron Radiation Facility. A monochromated X-ray beam with a photon energy of 10 keV (bandwidth <2 × 10$^{-4}$) was used. The beam size was cut to ~70 (horizontal) × 200 (vertical) μm$^2$ using a pair of motorized slits, leading to a footprint of ~225 × 200 μm$^2$ on the sample surface for the Bragg angle (~18.7°) of LSAT 002 reflection. This beam footprint fully covered the capacitor test structures with a diameter of 100 μm. The superlattices were mounted in the vertical scattering geometry on a four-circle diffractometer, and a tungsten probe station was attached on the latter to bias the samples. A 2D photon-counting area detector (Eiger 500 K, Dectris) was used to record the diffraction intensity. The 3D RSM results were reconstructed from raw measured data using custom Matlab and Python codes.

### Second-harmonic generation (SHG)
SHG measurements were performed on a home-developed laser-scanning microscope system[39]. A Ti:sapphire femtosecond laser (MaiTai, Spectra-Physics) was employed as the 1ω excitation source, emitting at 35 fs pulse width, 800 nm center wavelength, and an average power of 50–200 mW. For the normal incidence geometry, the 1ω laser beam was focused using an objective lens (numerical aperture = 0.55), which also collected the backscattered 2ω light signals. The excitation light polarization was controlled with a zero-order half waveplate, and the 2ω light polarization was analyzed using a Glan–Taylor prism polarizer. The 2ω light signals were bandpass filtered and recorded with a photomultiplier tube detector. For the 45° incidence geometry, the 1ω laser beam was directed on the

sample surface with a convex lens (numerical aperture ~0.5) via another optical path, and the samples were 45° tilted so that the upright reflected $2\omega$ light signals could be collected and detected using the same path as the backscattering geometry. In both cases, sharp tungsten probes were used to bias the 100-μm diameter capacitors in situ under SHG measurement. Electrical test signals were supplied by a functional generator (Rigol DG2052) or a source meter (Keithley 2470) which could measure the leakage current simultaneously. A heating stage was used to heat the samples in open air up to 450 K.

**Phase-field simulation**

Phase-field simulations were employed to investigate the polar structure of the $(PbTiO_3)_{14}/(SrTiO_3)_{16}$ (PTO/STO) superlattice on a LSAT substrate. The polarization evolution processes were obtained by solving the time-dependent Ginzburg–Landau equations:

$$\frac{\partial \mathbf{P}}{\partial t} = -L \frac{\delta F(\mathbf{P}, \nabla \mathbf{P})}{\delta \mathbf{P}} \tag{1}$$

where $\mathbf{P}$, $t$, $L$ are the spontaneous polarization vector, evolution timestep and kinetic coefficient, respectively. The total energy $F$ can be written as the volume integration of the Landau, mechanical, electrostatic, and polarization gradient energy densities, i.e.,

$$F = \int \left( f_{Land} + f_{elas} + f_{elec} + f_{grad} \right) dV \tag{2}$$

Detailed expressions for the individual energy density functionals[60] and the materials parameters of PTO and STO[43,52] can be found in previous reports.

A three-dimensional mesh of $192 \times 192 \times 350$ was used, with each grid representing 1 unit cell. The periodic boundary condition was applied along the $x$ and $y$ dimensions, while a superposition method was used in the out-of-plane direction[61]. Along the latter direction, the thicknesses of the substrate, thin film, and air were set as 30, 300, and 20, respectively. To account for the elastic inhomogeneity while solving the elastic equilibrium equation, an iterative perturbation method was used. Thin-film mechanical boundary conditions were adopted, where the out-of-plane stress components were fixed to zero on the top film surface, while on the substrate bottom sufficiently far away from the electrode/superlattice film interface, the out-of-plane displacement was set to be zero.

**Reporting summary**

Further information on research design is available in the Nature Portfolio Reporting Summary linked to this article.

# Data availability

The experimental and modeling data that support the findings of this study are available from the corresponding authors upon request.

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

## Acknowledgements

The authors thank Lingfei Wang and Jingdi Lu for providing the $NaIO_4$ etchant recipe and Jingbo Sun for useful discussion. This work was financially supported by Basic Science Center Project of the National Natural Science Foundation of China (NSFC) under grant No. 52388201 (Q.L., J.-F.L. and C.-W.N.), NSFC grant No. 52150092 (Q.L.), No. 52073155 (Q.L.), No. 92166104 (Z.H.), the Joint Funds of the NSFC grant No. U21A2067 (Z.H.), and by National Key R&D Program of China under Grant No. 2020YFA0309100 (Q.L.) and 2021YFA1202801 (Q.Z.), Beijing Municipal Natural Science Foundation under Grant No. 1212016 (Q.Z.). The in situ RSM experiment was carried out at the beamline BL02U2, Shanghai Synchrotron Radiation Facility (SSRF). The phase-field simulations are performed using the Mu-PRO software package (https://muprosoftware.com), on the MoFang III cluster on Shanghai Supercomputing Center (SSC).

## Author contributions

Q.L. conceived and designed the project. S.W. performed film growth and device fabrication. S.W., W.L. and C.D. performed SHG studies. S.W., W.L., Y.G. and X.L. performed in situ synchrotron XRD. H.G. and Q.Z. performed STEM imaging. Z.H. and Y.W. performed phase-field simulations. S.W., Z.H. and Q.L. analyzed the phase-field results. W.L. performed finite-element device modeling. P.G.E., J.-F.L. and C.-W.N. discussed the results. S.W. and Q.L. wrote the manuscript with input from all the authors.

## Competing interests

The authors declare no competing interests.
