## [Peer Review File · Nature Communications]

Giant electric field-induced second harmonic generation in polar skyrmionsReviewer #1 (Remarks to the Author):

Wang et al. investigated electric field-induced SHG in skyrmions, based on oxide superlattices. They claim achieving giant effect and propose potential application devices. The experimental investigation and theoretical simulation are comprehensive. The conclusions bare some promising gradients. However, the presentation is a bit less transparent. The really ground-breaking innovation point is not clearly identified. And some of the claims are not clearly underlined by experimental evidences. I have the following comments:

(1) I am not clear what is the key innovation of this work. The work is good or even excellent. However, it seems every technical elements it encompasses are discovered by previous works. What is the most unique aspect of this work? Please remind me if I missed any very pivotal points.

(2) This is an electrically-controlled proof-of-principle device. It can not work at an optical frequency. Hence, it is a boost to electronic or optoelectronic device, but hardly a potential photonics or optics platform. For example, this cannot be used for all-optical switching, and cannot achieve the ultrahigh speeds of all-optical switches.

(3) In Fig. 1e, the yellow arrows are an over-strong eye guidance to the readers. If removing the many arrows, I doubt there can be skyrmions identified. Fig. 1f is also of low contrast ration. Related to this, what does the authors mean (any figure illustration) of "polar" in "polar skyrmions"?

(4) The authors did not show how the TEM images of skyrmions change with external electric field. They only interpret the skyrmions are the cause to the SHG. "The applied out-of-plane electric field expands or shrinks the skyrmions, displacing their wall positions" is only a guess.

(5) Related to (4). Actually, the electric field can potentially modify the lattice structure, which can lead to the change in SHG. The statement "at the atomistic level, the SHG susceptibility is rooted in the anharmonicity of electronic structure" is incorrect. Lattice can easily modify the SHG. Spin or orbital can also. "skyrmion walls modulates the electronic structure with large anharmonicity" is a wrong statement. They neither discuss the role of lattice structure, nor give direct experimental evidence! Most are guesses and "possible" interpretations. It lacks firm experimental evidences.

(6) The skyrmions are at nanometer scale, it seems they did not consider/discuss subwavelength effects (e.g. Scientific Reports 3, 2358 (2013); Chin. Phys. Lett. 36, 027801 (2019)). The functioning of the superlattice is not discussed. The surface and interfaces in the superlattice surely play a role [Nano Energy 82 (2021) 105752].

(7) Some errors and blurry phrases. (A) SHG is about broken inversion symmetry, not anharmonicity. (B) Why say it is "giant"? (C) What does a previously hidden order parameter mean? (D) Why inter-skyrmion correlation si strong? (E) Why say anomalously large? Any evidence? (F) The expectation to GHz can be removed, because there is no experimental demonstration. (G) "Topological" is used frequently. It means no more than tomographic. It is a bit misleading to the readers. It has nothing to do with topological physics.

(8) In Fig. 4d, there are not many oxide superlattices cited. They may want to include [Nano Energy 82 (2021) 105752].

(9) Although the Fig. 4e,f is marvelous, I am not confident that the design can possibly be realized in the near future. There are many conventional ways to electrically control the light beams (including second harmonic waves). What is the advantage here? Do the authors really want to include this proposal? Why don't they realize it experimentally and submit another paper?

Minors:

There are some grammatical errors, such as "measurements... provides", "Fig. 2a" should be "Figure 2a", "Fig. 4b" should be "Figure 4b".

Reviewer #2 (Remarks to the Author):

In the paper "Giant electric field-induced second harmonic generation in polar skyrmions", Wang et al present an effective strategy to tuning the EFISH performance based on the polar skyrmions, a topological phase spontaneously formed in $\text{PbTiO}_3/\text{SrTiO}_3$ ferroelectric superlattice. They also establish the microscopic links between the exotic polarization configuration and field-induced transition paths of the skyrmions and the related EFISH response. I think this paper is well organized, with the experimental methods suitably used to study the microscopic mechanism. I would like to recommend publication on Nature Communications after the authors address my following questions.

1. I notice that the authors choose a ~ 96 nm $[(\text{PbTiO}_3)_{14}/(\text{SrTiO}_3)_{16}]_8$ superlattices as a model system. How are the performance of the other PTO/STO superlattices with other thicknesses?
2. The authors used phase field simulations to establish the link between the polarization configuration and field-induced transition paths of the skyrmions and the related EFISH response. If experimentally accessible, it will be helpful if they can visualize the structure by STEM before and after the switching (or in-situ TEM observations) to directly compare the polarization vectors to demonstrate the skyrmions transition. Or they can perform I-V measurements using CAFM to demonstrate the polarization transition process.
3. The authors state that "Such difference stems from the upward P_z direction of the skyrmion center in the pristine state, which is due to the epitaxial growth conditions. We further speculate that the initial state can be selected by modifying the growth conditions". I am wondering whether the direction of the skyrmion center in the pristine state can be tuned. And how to confirm the pristine state?
4. It was demonstrated that external electric field induced domain evolution, engenders a strong SHG response. Can the typical ferroelectric oxides such as PTO, BFO, exhibit EFISH? What is the advantage of PTO/STO superlattices over the other ferroelectric system?

Reviewer #3 (Remarks to the Author):

Reviewer information

The manuscript reports on the polar skyrmions formed in $\text{PbTiO}_3/\text{SrTiO}_3$ ferroelectric superlattices, exhibiting an outstanding comprehensive electric field-induced second harmonic generation (EFISH) performance. The susceptibility and modulation depth reach ~ 54.2 pm V^{-1} and $\sim 664\%$ V^{-1} , respectively. The in-situ experimental results and phase-field simulations are discussed and they found that the field-induced transition paths of the skyrmions and their EFISH response are related to the polarization configuration. The authors declare that the polar skyrmions cause the EFISH effect through a delicate, skyrmion wall-involved mechanism. The experimental results presented are very interesting. However, I have some concerns about the interpretation of the results and in particular the microscopic mechanisms invoked.

Comments: 1. The stacking structure. Why the authors chose the stacking structure $[(\text{PbTiO}_3)_{14}/(\text{SrTiO}_3)_{16}]_8$? Many structures can generate polar skyrmions. Have the authors tried any other stacking structures? In In 346, 'For instance, using a more tensile substrate of SrTiO_3 may stiffen the skyrmionic interaction and decline the EFISH modulation slope; reducing the SrTiO_3 component layer thickness may directly boost the global efficiency due to increased PbTiO_3 volume fraction'. Why does changing the substrate and changing the stacking structure cause a change in the EFISH response, is this change due to the change in ferroelectric domain structure?

Comments: 2. The defects. Are the properties of the superlattice sensitive to the defects? Such as oxygen vacancy or edge dislocations? It is clear the samples prepared by the authors have high quality. However, even for low misfits, there will be a certain critical thickness above which the formation of dislocations will take place to relax the strain. Will the defects change the skyrmion domain structure or affect the EFISH signals?

Comments: 3. The relaxation effect. In In 104, 'substrate misfit strain is partially (by about 1%, biaxially) relaxed within the SrRuO₃' Did the author take into consideration the relaxation effect? Strain relaxation may not always be homogeneous throughout the thickness of the superlattice structure. What is the stress effect on the upper surface compared with the freestanding one? The skyrmion structure may change and may affect the EFISH response. The relaxation state is also sensitive to the time, did the author re-examine the domain structure or the EFISH response after days or weeks?

Comments: 4. In the temperature- SHG intensity plots (Fig 4a), the authors should explain that when the temperature is above $\sim 370\text{K}$, why the value of SHG intensity of I (-10V) is greater than that of I (+10V), is this change of intensity related to the polarization configuration of the superlattice?

Comments: 5. It is a little bit difficult to read Fig 4f, where is the position of the Fresnel lens? What is the direction of the input light? I can guess them from the picture, but it is better to point them out for the readers. I didn't see any descriptions or details about the finite-element modeling in the manuscript or supplementary documents, are the authors using commercial software or certain programs to achieve the light simulation results?

Response to the reviewers' comments

Manuscript ID: NCOMMS-23-49620

Manuscript Title: Giant electric field-induced second harmonic generation in polar skyrmions

1. Reviewer #1 (Remarks to the Author):

Wang et al. investigated electric field-induced SHG in skyrmions, based on oxide superlattices. They claim achieving giant effect and propose potential application devices. The experimental investigation and theoretical simulation are comprehensive. The conclusions bare some promising gradients. However, the presentation is a bit less transparent. The really ground-breaking innovation point is not clearly identified. And some of the claims are not clearly underlined by experimental evidences. I have the following comments:

We truly appreciate the reviewer for providing a largely positive assessment as well as constructive comments.

1.1 Reviewer #1's Comment 1:

I am not clear what is the key innovation of this work. The work is good or even excellent. However, it seems every technical elements it encompasses are discovered by previous works. What is the most unique aspect of this work? Please remind me if I missed any very pivotal points.

Authors' Response:

Thank you for the affirmative comment about our work. Although polar skyrmions have been reported in a few studies, to the best of our knowledge, our work is the first to discover electrically tunable nonlinear optical properties of topological polar structures. The polar skyrmion structure holds significantly higher EFISH modulation depth than other materials reported and considerable $\chi^{(2)}$ (shown in **Fig. 4d**), offering new opportunities for discovering and designing optoelectronic functionalities. We also elucidate the mechanisms of the observed EFISH modulation, deeply connecting the nonlinear optical properties with the microscopic structures (specifically, electric field-induced evolutions of the polarization configuration) in the skyrmions. Such a topic had not been tackled before and the understanding was completely lacking. Therefore, we

believe that our work not only opens new pathways for the applications of topological
domain structures (including polar skyrmions, polar vortices, etc.), but also enlightens
further discoveries of emerging nonlinear optical properties of novel ferroelectric
materials.

**1.2 Reviewer #1's Comment 2:**

This is an electrically-controlled proof-of-principle device. It can not work at an
optical frequency. Hence, it is a boost to electronic or optoelectronic device, but hardly
a potential photonics or optics platform. For example, this cannot be used for all-optical
switching, and cannot achieve the ultrahigh speeds of all-optical switches.

**Authors' Response:**

Thanks for offering this viewpoint. We loosely used the word “platform”, and
would like to tone down by removing this word in the revised manuscript. As we have
demonstrated the skyrmionic EFISH effect, the superlattices can certainly be used for
electro-optic modulators and nonlinear optic devices, and compared to LNOI platform
(which has already received more than a decade’s research), they are advantageous in
the electric field tunability (LNOI is not so tunable because of the exceedingly large
coercive electric fields of LiNbO_3). As regards all-optical switching (recently realized
on LNOI platform), it is fundamentally based on nonlinear optic properties of materials
and could also be explored for polar skyrmions in future studies.

Please see **Abstract** “Our study not only presents a highly competitive thin-film
material (~~platform~~) ready for constructing on-chip devices...” and the last paragraph in
**Page 12** “The form of thin-film superlattices presents multiple degrees of freedom for
further tuning of the EFISH properties, in addition to an integratable material
(~~platform~~).” in the revised manuscript.

**1.3 Reviewer #1's Comment 3:**

In Fig. 1e, the yellow arrows are an over-strong eye guidance to the readers. If
removing the many arrows, I doubt there can be skyrmions identified. Fig. 1f is also of
low contrast ration. Related to this, what does the authors mean (any figure illustration)
of “polar” in “polar skyrmions” ?

**Authors' Response:**

We appreciate your valuable concerns. “Polar” literally means “related to dipoles
or polarization” (see the original definitions of polar skyrmion and polar vortex in
Nature 530, 198 (2016) and Nature 568, 368 (2019)). The usage of “polar” is also
counterpart to magnetic skyrmions or vortices which are formed by spins rather than
electric dipoles.

The yellow arrows mark the direction and magnitude of local polarization dipoles
for each unit cell, which are calculated from the locations of Pb/Sr/Ti ions based on the
measured HAADF-STEM images (not simply eye guidance). As the polar ion
displacements in ferroelectrics are rather small (up to a few 10s pm, only resolvable
using aberration-corrected S/TEM), one cannot visualize polar skyrmions or other polar
structures directly from the STEM images, but relies on the reconstructed dipole maps.
This analytical and presenting method has been previously adopted in a large number
of studies. The contrast of **Fig. 1f** is indeed somewhat low, because it is a vectorial map
of local electric field; in principle, it should be slowly varying across the skyrmions
compared with the dipole maps in **Fig. 1e**. While the measurement quality could be
marginally improved, **Fig. 1f** clearly reveals the correlated variations of the polarization
and electric field, as shown in **Fig. 1g**.

**1.4 Reviewer #1’s Comment 4:**

The authors did not show how the TEM images of skyrmions change with external
electric field. They only interpret the skyrmions are the cause to the SHG. “The applied
out-of-plane electric field expands or shrinks the skyrmions, displacing their wall
positions” is only a guess.

**Authors’ Response:**

As the reviewer would agree, spectroscopic methods have often been applied to
reveal motions of matter without directly providing real-space, real-time visualizations.
In this study, based on the systematic EFISH measurements, *in-situ* synchrotron X-ray
diffraction (which has provided microscopic evidences from the measured lattice strain
and satellite peak intensities) and phase-field simulations, we consider that our analysis
of the polarization transition process of the skyrmions is self-consistent and sufficient.
The skyrmion wall displacement has also been described in a previous study (Nature
Materials 20, 194 (2021)), where researchers have confirmed the existence of skyrmion

shrinking and wall displacement through a combination of phase-field simulation, XRD
and time-dependent dielectric measurements.

We do share the reviewer's curiosity to *in-situ* observe the electric field induced
structure changes of the skyrmions using Cs-corrected STEM (note that both *in-situ*
electric field and sub-angstrom resolution are required to achieve such a goal).
Unfortunately, the experimental challenges went beyond our initial expectation. We
managed to prepare three specimens using FIB, but they all suffered an early (below ~5
109 V) electrical breakdown starting from the SrRuO₃ bottom electrode layer and thus were
110 destroyed under the STEM, as depicted in **Fig. R1**. Note that we had to polish the
111 specimens very thin using FIB in order to obtain high-quality HAADF images and
112 thereby resolve the skyrmions; the 5 nm SrRuO₃ layer may thus become significantly
degraded. We infer that a possible solution is to replace the SrRuO₃ bottom electrode
layer with a conducting substrate (such as Nb:SrTiO₃) to avoid the electrical weak point.
This then will involve more effort to grow superlattices on a new type of substrate. In
any case, an *in-situ* STEM study of the skyrmions is significant and once fulfilled, the
results can be reported in a separate article (see e.g. Phys. Rev. Lett. 129, 107601
(2022)).

**Figure R1.** HAADF-STEM image of a superlattice specimen after electrical
breakdown.

**1.5 Reviewer #1's Comment 5:**

Related to (4). Actually, the electric field can potentially modify the lattice
structure, which can lead to the change in SHG. The statement “at the atomistic level,
the SHG susceptibility is rooted in the anharmonicity of electronic structure” is
incorrect. Lattice can easily modify the SHG. Spin or orbital can also. “skyrmion walls

modulates the electronic structure with large anharmonicity” is a wrong statement.
They neither discuss the role of lattice structure, nor give direct experimental evidence!
Most are guesses and “possible” interpretations. It lacks firm experimental evidences.

**Authors’ Response:**

We have re-studied the fundamentals of nonlinear optics. In noncentrosymmetric
materials, the potential energy function of electrons can be expressed as:

$$U(\tilde{x}) = \frac{1}{2}m\omega_0^2\tilde{x}^2 + \frac{1}{3}ma\tilde{x}^3$$

where \tilde{x} is the electron position, a is a parameter that characterizes the strength of
nonlinearity, ω_0 is the resonance frequency, and m is the mass of electrons. The third-
order term $\frac{1}{3}ma\tilde{x}^3$ is an anharmonic correction, which is essentially zero in
centrosymmetric systems where $U(\tilde{x}) = U(-\tilde{x})$. This anharmonicity results in a
potential well that is not perfectly parabolic, thus leading to the anharmonic oscillation
of the electrons and the generation of the SHG light. By solving the equation of electron
motion, we can derive the second-order nonlinear susceptibility $\chi^{(2)}$:

$$\chi^{(2)}(2\omega, \omega, \omega) = \frac{N(e^3/m^2)a}{\epsilon_0 D(2\omega)D^2(\omega)}$$

where ω is the frequency of the fundamental wave, $D(\omega) = \omega_0^2 - \omega^2 - 2i\omega\gamma$ is the
complex denominator function (γ the damping factor of the oscillation of electrons),
ϵ_0 is the permittivity of free space, $-e$ is the electron charge, and N is the number
density of atoms. $\chi^{(2)}$ follows a proportional relationship with the parameter a , which is
defined in the anharmonic correction term as discussed above. Therefore, we conclude
that SHG originates from the anharmonicity of electronic structure. (See Boyd and
Robert W. *Nonlinear Optics (3rd Edition)*. Elsevier. 2008 for more details.)

As the reviewer points out, lattice structure, spin and orbital can all influence the
SHG properties. In fact, they do so by affecting electronic potential energy distributions
and introducing anharmonicity in the electronic structure. Therefore, the SHG
susceptibility fundamentally roots from the anharmonicity of electronic structure,
which in turn reflects the changes in lattice structure, spin and orbital (the latter two can
be neglected for the polar skyrmion system).

Contrary to the reviewer’s comment, our study has been focused on the lattice
structure based on experimental observations and theoretical modeling. From *in-situ*
synchrotron XRD results, the 00L diffraction peaks at lower Q_z observed at ± 15 V

(shown in **Fig. 3b**) are caused by the larger portion of c -domain under external fields,
which evidences a change in the lattice structure. This change is accompanied by
modulation in the satellite peak intensities which evidences movement of the skyrmion
walls. The latter process leads to the breaking of the pseudo-centrosymmetry, as
faithfully reproduced by the phase-field modeling (see **Fig. 3e**). In summary, the
symmetry-breaking caused by the skyrmion wall movement and larger c/a in the c -
domains introduce larger anharmonicity in the electronic structures, which can be
experimentally verified by the XRD RSM and SHG results.

Setting those detailed arguments aside, we believe that here the lattice structure
change, the symmetry-breaking and the anharmonicity of electronic structure are just
different levels of description for the same physical process going on in the skyrmions.

**1.6 Reviewer #1's Comment 6:**

The skyrmions are at nanometer scale, it seems they did not consider/discuss
subwavelength effects (e.g. Scientific Reports 3, 2358 (2013); Chin. Phys. Lett. 36,
027801 (2019)). The functioning of the superlattice is not discussed. The surface and
interfaces in the superlattice surely play a role [Nano Energy 82 (2021) 105752].

**Authors' Response:**

Thank you for pointing out these factors. The $\text{PbTiO}_3/\text{SrTiO}_3$ superlattice system
is significantly different from $\text{LaTiO}_3/\text{SrTiO}_3$, because PbTiO_3 layers are ferroelectric.
As a matter of fact, the structure of the superlattice (including surfaces and interfaces)
does not change under external voltage, while the skyrmion structure (nanoscale
domain) changes. Therefore, the observed EFISH response and SHG polarimetry
results as well as *in-situ* X-ray diffraction results cannot be related to the surfaces and
interfaces. The superlattice here plays the fundamental role of hosting the existence and
evolution of polar skyrmions, which has been thoroughly discussed in the manuscript.

Regarding the subwavelength effects associated with Fabry–Pérot resonances
(Scientific Reports 3, 2358 (2013); Chin. Phys. Lett. 36, 027801 (2019)), the scenario
of polar skyrmions can be significantly different since their size are far smaller than the
probing wavelength (~ 8 nm vs. 800 nm), which also goes beyond the theoretical
framework of Mie resonance or plasmonic resonance that occur at the subwavelength
scale. Therefore, we believe that the skyrmions collectively should be regarded as a
single phase, due to such a two-order of magnitude difference in the length scales.

We truly appreciate the reviewer for bringing these interesting and enriching
papers to our attention and have cited these papers in the revised manuscript.
Nevertheless, we have refrained from making excessive discussions against all the
plausible mechanisms or effects (neither could we exhaust them), lest the main ideas
are diluted and the manuscript is over complicated. We have provided a brief comment
regarding these extrinsic mechanisms in the revised manuscript **Page 10** “We further
point out that other extrinsic mechanisms, such as subwavelength structures^{47,48} and
superlattice interfacial effects⁴⁹, can be ruled out here based on our observations.”

**1.7 Reviewer #1’s Comment 7:**

Some errors and blurry phrases. (A) SHG is about broken inversion symmetry, not
anharmonicity. (B) Why say it is “giant” ? (C) What does a previously hidden order
parameter mean? (D) Why inter-skyrmion correlation is strong? (E) Why say
anomalously large? Any evidence? (F) The expectation to GHz can be removed,
because there is no experimental demonstration. (G) “Topological” is used frequently.
It means no more than tomographic. It is a bit misleading to the readers. It has nothing
to do with topological physics.

**Authors’ Response:**

Thank you for offering these detailed comments, and we would like to respond
point-to-point in the following:

(A) Please see **Response 1.5**, the electronic potential energy anharmonicity and
the broken inversion symmetry of lattice structure and polarization distribution are two
sides of the same phenomenon. Therefore, the breaking of the pseudo-centrosymmetry
caused by the skyrmion wall movement, as described in “Structural mechanisms of the
EFISH effect”, is also a process of introducing anharmonicity, leading to the enhanced
SHG response.

(B) The word “giant” here refers to the EFISH modulation depth of our
superlattices, which is significantly larger than common materials. Compared to
previously reported materials/devices with an EFISH modulation depth of $\sim 1-10\% V^{-1}$,
$(PbTiO_3)_{14}/(SrTiO_3)_{16}$ superlattices ($664\% V^{-1}$) holds near 100 times the value for most
materials, and close to twice of the second highest materials (PFO polymers), as shown
in **Fig. 4d**. Besides, the induced $\chi^{(2)}$ is also considerably large (larger than LNOI and

LaTiO₃/SrTiO₃ superlattice), only lower than those 2D materials resonantly enhanced
at electronic transitions.

(C) As shown in **Fig. 3f**, the profile of the inter-skyrmion correlation function (at
$\sim 0.10 \text{ nm}^{-1}$) reproduces the EFISH intensity trends. Therefore, the inter-skyrmion
correlation function can be an order parameter for describing the degree of pseudo-
symmetry breaking caused by the shrinking or expanding of polar skyrmions. Usually,
the polarization and topological charge number are regarded as the order parameters of
topological polar structure systems.

(D) The inter-skyrmion correlation here is the spatial correlation of the walls of the
closest pair of skyrmions, and originates from the arrangement and size of skyrmions.
Such correlation distance is the distance between each pair of skyrmions. The strong
correlation means that the size of and distance between the skyrmions are similar across
different skyrmions, and that the distance between the opposite walls of a single
skyrmion does not equal to the distance between the closest pair of walls of neighboring
skyrmions (otherwise the intra-skyrmion correlation will be in the leading place), which
marks the broken pseudo-centrosymmetry and large SHG intensity. Our RSM results
(**Fig. 1b** and **3a**) have also shown the periodicity or spatial correlation of skyrmions,
which is consistent with results from previous articles [Nature 568, 368 (2019); Nature
Materials 20, 194 (2021)].

(E) The overall $\chi_{33}^{(2)}$ is larger at positive voltages than negative voltages, which is
measured by experiment in **Fig. 2** and calculated in “EFISH response of the polar
skyrmions” section. The calculated maximum $|\chi_{33}|$ is $\sim 54.2 \text{ pm V}^{-1}$ in the PbTiO₃ layers
of the superlattices, which is about three times of the bulk value ($\sim 17 \text{ pm V}^{-1}$) of PbTiO₃.
We attribute this difference on the higher skyrmion wall density at positive voltages;
therefore we conclude that the skyrmion domain walls obtain large $\chi_{33}^{(2)}$.

(F) As we have experimentally demonstrated, the EFISH response speed is very
high ($>10 \text{ MHz}$), and no significant decrease of EFISH intensity has been observed at
10 MHz , implying the possibilities to further increase the working frequency. We are
intrigued to explore the modulation with electric field frequency up to GHz scale.
However, as the modulation signal steps into microwave frequencies, the shape and
structure of the electrodes and peripheral circuits should be carefully designed, which
is difficult for the current capacitor devices. Therefore, we make a prediction of the
EFISH operation at GHz frequencies in polar skyrmions, and will try to experimentally
achieve it in further studies. For now, it is true that such prediction lacks a solid

experimental proof, so we have changed the description of the expectation to GHz into
“...high response bandwidth (higher than 10 MHz)...” in the Abstract of the revise
manuscript.

(G) The word ‘Topological’ is not misused or a misconception here. Topological
physics includes two types topologies in both reciprocal (momentum space, as in
topological insulators, Weyl semimetals, etc.) and real space. The latter is of no less
importance, and has been long and extensively addressed in magnetic and ferroelectric
topological structures, liquid crystals and other material systems (see e.g. P. M. Chaikin
and T. C. Lubensky, *Principles of Condensed Matter Physics*. Cambridge University
Press 1995). For our current study, a very comprehensive account of the relevant
physics can be found in a recent review article (Junquera et al, *Topological phases in*
*polar oxide nanostructures*. Rev. Mod. Phys. 95, 025001 (2023))

In fact, our results show that the phase transition from polar skyrmions to *c*-domain
involves changes in the Pontryagin density (as shown in **Fig. 3d**), or topological charge
of the system. Therefore, it is a kind of topological phase transition. In polar skyrmions,
the topological charge value, which is the surface integration of Pontryagin density, is
± 1 , depending on the polarization direction of skyrmion center (see also Nature 568,
368 (2019); Nature Materials 20, 194 (2021); Nature Communications 14, 1355 (2023)].
The robust and high EFISH performance of the skyrmions are closely associated with
their topologically protected phase stabilities under external electric field, which are
absent in conventional ferroelectric materials with domain walls.

**1.8 Reviewer #1’s Comment 8:**

In Fig. 4d, there are not many oxide superlattices cited. They may want to include
[Nano Energy 82 (2021) 105752].

**Authors’ Response:**

Thank you for the advice. We have modified **Fig. 4d** to include the LaTiO₃/SrTiO₃
superlattice reported in this paper. Since this oxide superlattice was not demonstrated
to show SHG modulation induced by external electric field, its modulation depth has
been taken as zero in **Fig. 4d**.

**1.9 Reviewer #1’s Comment 9:**

Although the Fig. 4e,f is marvelous, I am not confident that the design can possibly
be realized in the near future. There are many conventional ways to electrically control
the light beams (including second harmonic waves). What is the advantage here? Do
the authors really want to include this proposal? Why don't they realize it
experimentally and submit another paper?

**Authors' Response:**

Thanks for the viewpoint. The advantage of this particular device example is that
it can provide extra high modulation depth at the focal point, giving a broadened range
of SHG intensity modulation. We envisage that the EFISH effect of the skrymions can
also be utilized for designing several other important nonlinear optic devices, including
quasi-phase matching waveguide and tunable metasurface.

The reviewer is correct that the nanofabrication of these devices is challenging (but
certainly feasible using EBL, RIE, FIB, etc.). While we are highly motivated to shift
our current research focus towards nanophotonic devices, there are still important
problems at the materials side to delve into (as being discussed here with all the
reviewers). For this reason, we hope to provide an introductory example to draw interest
from the nanophotonics communities; in fact, the Reviewer-3 raised useful comments
about the device design (see 3.5).

That being said, we would be further open to the reviewers' suggestions and can
make adjustment in the presented results as appropriate.

**1.10 Reviewer #1's Comment 10:**

Minors: There are some grammatical errors, such as "measurements... provides",
"Fig. 2a" should be "Figure 2a", "Fig. 4b" should be "Figure 4b".

**Authors' Response:**

Thank you for pointing out these errors. We have carefully polished the manuscript
and corrected these errors. The "Fig." abbreviation is a common format style for *Nature*
*Communications*. Therefore, we have kept the use of this abbreviation.

**2. Reviewer #2 (Remarks to the Author):**

In the paper “Giant electric field-induced second harmonic generation in polar
skyrmions”, Wang et al present an effective strategy to tuning the EFISH performance
based on the polar skyrmions, a topological phase spontaneously formed in PbTiO₃/
SrTiO₃ ferroelectric superlattice. They also establish the microscopic links between
the exotic polarization configuration and field-induced transition paths of the skyrmions
and the related EFISH response. I think this paper is well organized, with the
experimental methods suitably used to study the microscopic mechanism. I would like
to recommend publication on Nature Communications after the authors address my
following questions.

We greatly appreciate the affirmative assessment and constructive comments of the
reviewer.

**2.1 Reviewer #2's Comment 1:**

I notice that the authors choose a ~96 nm [(PbTiO₃)₁₄/(SrTiO₃)₁₆]₈ superlattices as
a model system. How are the performance of the other PTO/STO superlattices with
other thicknesses?

**Authors' Response:**

Thank you for your valuable question. In the original manuscript, we discuss the
potential influence of the layer thickness on the EFISH performance. Overall, we
envisage that the layer thickness can change the stability of polar skyrmions (in extreme
cases, the skyrmions disappear) and modify the EFISH performance. To experimentally
demonstrate it, we have grown another superlattice with thinner PbTiO₃ layers,
[(PbTiO₃)₁₁/(SrTiO₃)₁₆]₈, and performed EFISH measurements together with phase-
field modeling, as presented in **Fig. R2**. This superlattice also exhibits strong EFISH
response. The measured maximum EFISH modulation depth is ~528% V⁻¹ at 7 V, and
the calculated maximum $|\chi_{33}|$ is ~52.8 pm V⁻¹ at 15 V, which are close to the
corresponding values of [(PTO)₁₄/(STO)₁₆]₈ (664% V⁻¹ and 54.2 pm V⁻¹). Interestingly,
this superlattice shows some unique characteristics; for example, the *p*-in/*p*-out and *s*-
in/*p*-out response are different at the negative bias side (cf. **Fig. R2a,b** and **Fig. 2** in the
manuscript). This can preliminarily be understood by the phase-field simulation results
(cf. **Fig. R2d** and **Fig. 3d**). In contrast to [(PTO)₁₄/(STO)₁₆]₈, the pristine state
skyrmions are more stripe-like due to modified stability conditions. Under applied

electric fields, the stripe-like skyrmions gradually break into round skyrmions with two
 opposite center polarization directions. The configuration at negative voltages
 corresponds to center-convergent skyrmions, compared to center-divergent skyrmions
 at positive voltages. Currently, we are still investigating more superlattices with
 different thickness and will report the results in a separate paper.

Figure R2. EFISH response of the polar skyrmions in $[(\text{PbTiO}_3)_{11}/(\text{SrTiO}_3)_{16}]_8$ superlattices measured using capacitor devices. **a**, Measured SHG intensity as a function of applied voltage for different polarization states (p -, parallel and s -, perpendicular to the plane of incidence) of the input fundamental light. The bias voltage sequence in each cycle was from 0 V to 15 V, then to -15 V and back to 0 V. **b,c**, Polar plots of the input polarization angle-dependent SHG intensity measured under **(b)** 0° (p -out) and **(c)** 90° (s -out) output polarization conditions. Lines are model fitting results to the measured data points. An average laser power of ~ 150 mW was delivered onto the samples in these measurements. **d**, Phase-field simulated in-plane polarization vector maps illustrating the evolution of the skyrmions within a bias range of ± 15.7 V.

**2.2 Reviewer #2's Comment 2:**

The authors used phase field simulations to establish the link between the
polarization configuration and field-induced transition paths of the skyrmions and the
related EFISH response. If experimentally accessible, it will be helpful if they can
visualize the structure by STEM before and after the switching (or in-situ TEM
observations) to directly compare the polarization vectors to demonstrate the skyrmions
transition. Or they can perform I-V measurements using CAFM to demonstrate the
polarization transition process.

**Authors' Response:**

Thanks for the suggestions. We have attempted to perform *in situ* STEM to directly
visualize the structural transitions of the skyrmions with applied electric field. Note that
*ex situ* observations cannot fulfil such a goal because the induced states disappear once
electric field is withdrawn. Unfortunately, we have not been successful so far because
the FIB-prepared specimens always suffered breakdown at low voltages (less than 5 V).
One of the failed results is shown in **Fig. R1** with explanation in **Response 1.4**.

Following the suggestions, we have also performed I-V and C-V measurements on
the capacitor devices, as presented in **Fig. R3**. It appears that the current density
measured under d.c. voltage is higher than and overwhelms the displacement current
that may occur during the polarization transition. Therefore, the skyrmion transition
process may not be observable in standard I-V measurements. On the other hand, the
measured C-V curves reveal a large decrease in capacitance under external voltages.
This large dielectric tunability is similar to a previous report (Nature Materials 20, 194
(2021)) and can also demonstrate the field-induced transitions of the skyrmions.

**Figure R3.** Macroscopic current-voltage (a) and capacitance-voltage (b) curves on
[(PbTiO₃)₁₄/(SrTiO₃)₁₆]₈ superlattices. The current was measured by a source meter

(Keithley 2470) on Al electrodes with a thickness of 65 nm. The capacitance of the
device was measured at 100 kHz using an impedance analyzer (Keysight E4990A) on
SrRuO₃ electrodes.

**2.3 Reviewer #2's Comment 3:**

The authors state that “Such difference stems from the upward Pz direction of the
skyrmion center in the pristine state, which is due to the epitaxial growth conditions.
We further speculate that the initial state can be selected by modifying the growth
conditions”. I am wondering whether the direction of the skyrmion center in the pristine
state can be tuned. And how to confirm the pristine state?

**Authors' Response:**

The initial polarization direction is related to the growth conditions of the films,
including oxygen pressure, growth temperature, interface chemical states and others, as
recently demonstrated for PZT thin films (Adv. Funct. Mater. 33, 2214849 (2023)). We
thus suggest that it can also be modified for PbTiO₃/SrTiO₃ superlattices through tuning
the growth condition. Nevertheless, it has to be pointed out that the tuning of suitable
growth conditions for superlattices can be more involved than single-layer films, and
the stability of polar skyrmions may also be altered due to changes in the strain states
and boundary conditions. This can be a meaningful direction in our future research.

The pristine state of the skyrmion center can be reliably identified using HAADF-
STEM or 4D-STEM, as illustrated in **Fig. 1** of this manuscript and in previous studies
(see e.g. Nature Materials 20, 194 (2021)). Because of the small size of skyrmions (~8
446 nm periodicity), it is very challenging to directly observe the skyrmion center with other
microscopy methods, rendering STEM the only effective tool to the knowledge of the
authors. However, since it has been demonstrated that the EFISH response is correlated
with the field-induced transition processes of the skyrmions, our study can provide a
complementary, optic spectroscopy method for determining their dipole configurations.
Apart from the nonlinear optical properties, we envisage that the microscopic states of
the skyrmions may potentially be reflected in other macroscopic properties.

**2.4 Reviewer #2's Comment 4:**

It was demonstrated that external electric field induced domain evolution,
engenders a strong SHG response. Can the typical ferroelectric oxides such as PTO,

BFO, exhibit EFISH? What is the advantage of PTO/STO superlattices over the other
ferroelectric system?

**Authors' Response:**

In our study, we found that typical single-layer thin films of PTO and STO cannot
exhibit EFISH responses under the same measurement conditions with PTO/STO
superlattices, as presented in Supplementary **Fig. S3**. In several parallel studies, we
have measured a number of other materials, such as sol-gel PZT films, KNN films and
PbZrO₃ films, and observed various phenomena: some of them show somewhat EFISH
response or changes in the SHG signals under electric field due to the induced domain
evolutions. The EFISH effect is not unique to PTO/STO superlattices per se.

We believe that the key advantage with polar skyrmions (and likely other types of
topological structures as well) lies in their characteristic nanoscale domain evolution
mechanisms. In conventional ferroelectric materials, domain switching usually happens
at much larger length scales and in far less coordinated manners, therefore forming
abrupt changes at certain coercive fields and large hysteresis in the SHG response (see
e.g. Nature 466, 954 (2010)). By contrast, polar skyrmions can collectively expand or
shrink at low to moderate external voltages before switching to a single *c*-domain at
large voltages; the resultant EFISH effect does not show abrupt changes nor obvious
hysteresis, enabling continuous SHG modulation with large dynamic range, high speed
and fatigue-resistance.

**3. Reviewer #3 (Remarks to the Author):**

The manuscript reports on the polar skyrmions formed in PbTiO₃/SrTiO₃
ferroelectric superlattices, exhibiting an outstanding comprehensive electric field-
induced second harmonic generation (EFISH) performance. The susceptibility and
modulation depth reach ~54.2 pm V⁻¹ and ~664% V⁻¹, respectively. The in-situ
experimental results and phase-field simulations are discussed and they found that the
field-induced transition paths of the skyrmions and their EFISH response are related to
the polarization configuration. The authors declare that the polar skyrmions cause the
EFISH effect through a delicate, skyrmion wall-involved mechanism. The experimental
results presented are very interesting. However, I have some concerns about the
interpretation of the results and in particular the microscopic mechanisms invoked.

We are deeply grateful for the reviewer's interest in this study and have attempted to
address her/his concerns in what follows:

**3.1 Reviewer #3's Comment 1:**

The stacking structure. Why the authors chose the stacking structure
[(PbTiO₃)₁₄/(SrTiO₃)₁₆]₈? Many structures can generate polar skyrmions. Have the
authors tried any other stacking structures? In In 346, 'For instance, using a more
tensile substrate of SrTiO₃ may stiffen the skyrmionic interaction and decline the
EFISH modulation slope; reducing the SrTiO₃ component layer thickness may directly
boost the global efficiency due to increased PbTiO₃ volume fraction'. Why does
changing the substrate and changing the stacking structure cause a change in the EFISH
response, is this change due to the change in ferroelectric domain structure?

**Authors' Response:**

The choice of this particular stacking structure, [(PbTiO₃)₁₄/(SrTiO₃)₁₆]₈, is
somewhat serendipitous; we first observed the EFISH phenomena in this superlattice
and then performed a systematic study based on it. We fully agree with the reviewer
that other stacking structures can also generate polar skyrmions, and make a discussion
about their potential influence on the EFISH response (as the reviewer quoted here).

Indeed, changing the substrate and/or stacking structure can change the stability of
the skyrmions (essentially nanoscale ferroelectric domains), which may influence their
phase transition conditions under applied electric field and accordingly modify the
EFISH response. More specifically, changing the substrate will change the strain

boundary conditions for the superlattice. When a more tensile substrate is adopted, the
tensile stress imposed on the superlattice can increase the tendency to form *a*-domains,
and it becomes more difficult to turn the in-plane polarization into the out-of-plane
directions. This may hinder the movement of skyrmion walls during the field induced
phase transitions, thus leading to a stiffened interaction between the skyrmions. On the
other hand, since the majority of the polarization of the superlattice occurs in PbTiO₃
layers (which are far more SHG active than SrTiO₃), the global SHG intensity increases
with the PbTiO₃ volume fraction for a certain total thickness of superlattice (but
certainly, the skyrmion structure has to be maintained to exhibit an EFISH effect).
Furthermore, we would like to comment that the functional responses of polar
skyrmions and other topological structures are closely related to their structure
stabilities and warrant more detailed investigations.

We thank the reviewer's advice to try other stacking structures. During the revision,
we have experimentally demonstrated the EFISH effect in another skyrmion-hosting
[(PbTiO₃)₁₁/(SrTiO₃)₁₆]₈ superlattice. This superlattice exhibits large EFISH response
yet with somewhat different characteristics. Please see the main results in **Fig. R2** and
relevant discussion there.

**3.2 Reviewer #3's Comment 2:**

The defects. Are the properties of the superlattice sensitive to the defects? Such as
oxygen vacancy or edge dislocations? It is clear the samples prepared by the authors
have high quality. However, even for low misfits, there will be a certain critical
thickness above which the formation of dislocations will take place to relax the strain.
Will the defects change the skyrmion domain structure or affect the EFISH signals?

**Authors' Response:**

For oxide superlattices, the presence of oxygen vacancies is dependent on the
growth conditions and usually unavoidable. In our superlattices, the edge dislocations
also occur in the SrRuO₃ layers to release the large misfit strain with LSAT substrates,
as directly observed from the HAADF image in **Fig. S1**. Nevertheless, we consider that
these defects altogether do not obviously affect the skyrmion domain structure and
EFISH signals, based on several experimental facts. 1) The measured EFISH response
speed is very high (>10 MHz). A defect-dominated process (such as migration of

oxygen vacancies) usually is much slower; 2) The good cycling fatigue behavior ($>10^{10}$
cycles), which is also beyond the usual range for defect-dominated processes.

While the EFISH properties appear not to be sensitive to the defects in our current
case, we stress that the defects still need to be controlled at certain levels to reach a
satisfactory epitaxial quality. We did grow a large number of inferior quality superlattice
samples that showed no or weak signs of the presence of skyrmions (thus not continued
for property measurements) before getting the ones presented in this manuscript.

**3.3 Reviewer #3's Comment 3:**

The relaxation effect. In ln 104, 'substrate misfit strain is partially (by about 1%,
biaxially) relaxed within the SrRuO₃' Did the author take into consideration the
relaxation effect? Strain relaxation may not always be homogeneous throughout the
thickness of the superlattice structure. What is the stress effect on the upper surface
compared with the freestanding one? The skyrmion structure may change and may
affect the EFISH response. The relaxation state is also sensitive to the time, did the
author re-examine the domain structure or the EFISH response after days or weeks?

**Authors' Response:**

The experimental results in the manuscript were obtained across a period of nearly
one year. For example, the synchrotron RSMs in **Fig. 1b** and **Fig. 3a-c** were measured
five months before the laboratory XRD RSM in **Fig. S1**; they show consistent satellite
peaks, indicating that the skyrmion domain structure are stable over the time period.
During the revision, we also repeated the EFISH measurements on the same superlattice
samples. All the obtained results do not show a time varying effect. This is because, as
we believe, after the film growth is finished at 650 °C, the partially relaxed strain state
is essentially frozen at room temperature.

Unlike the SrRuO₃ buffer layer in which the misfit strain is partially released, the
superlattice structure exhibits a homogeneous in-plane strain state throughout the
thickness. As illustrated by the 013 and 103 RSMs in **Fig. S1**, the Q_x of the superlattice
peaks are all of the same value with an offset to that of LSAT. The in-plane strain
homogeneity is also reflected by the well-defined, multiple-order superlattice peaks
along the Q_z direction, since the in-plane and out-of-plane lattice parameters are
correlated (lower-quality superlattice samples show very smeared superlattice peaks
and in those samples, the in-plane strain state can be inhomogeneous). As the in-plane

lattice parameters of the superlattice are close to SrTiO₃ (since SrTiO₃ is a constituent
layer of the superlattice), we describe it as a partially relaxed state (as if the superlattice
was grown on SrTiO₃ substrates). However, the superlattice is not fully relaxed as is
the case of free-standing films. **Fig. R4** illustrates the thickness variations of the in-
plane lattice parameters are rather minimal.

Figure R4. Thickness variations of the average in-plane lattice parameters of the top PbTiO₃ and SrTiO₃ layers

3.4 Reviewer #3's Comment 4:

In the temperature- SHG intensity plots (Fig 4a), the authors should explain that when the temperature is above ~370K, why the value of SHG intensity of I (-10V) is greater than that of I (+10V), is this change of intensity related to the polarization configuration of the superlattice?

Authors' Response:

Thank you for raising this critical problem. We have repeated the temperature-dependent EFISH measurements on the same superlattice sample, and confirmed that the higher intensity of I(-10 V) was more of a coincidence. **Fig. R5** presents three typical sets of results measured on different electrodes using identical conditions, showing that in most cases I(-10 V) should be smaller than I(+10 V). Such a subtle difference may suggest the presence of small structural variations over a large film area (10 × 10 mm²). It should be related to the polarization configuration of the superlattice, and possibly can be confirmed using X-ray microdiffraction techniques in future studies.

In the revised manuscript, we have replaced the original **Fig. 4a** with the results in
**Fig. R5b** to report more representative behaviors.

**Figure R5.** Temperature dependent SHG intensities measured at applied voltages of
10/-10 V under the *p*-in/*p*-out conditions, at multiple electrodes on
$[(\text{PbTiO}_3)_{14}/(\text{SrTiO}_3)_{16}]_8$ superlattice.

**3.5 Reviewer #3's Comment 5:**

It is a little bit difficult to read Fig 4f, where is the position of the Fresnel lens?
What is the direction of the input light? I can guess them from the picture, but it is better
to point them out for the readers. I didn't see any descriptions or details about the
finite-element modeling in the manuscript or supplementary documents, are the authors
using commercial software or certain programs to achieve the light simulation results?

**Authors' Response:**

Thanks for the suggestions. We have now provided a description of the simulation
in **Supplementary Note 5**. The simulation is performed using COMSOL Multiphysics
software. The lens is marked as the 45° line and the input light is incident from the left
side of the graph in the model, which is equivalent with the case when the input light is
incident from the top side and reflected to the right according to the reflection symmetry.
To clarify this, we have added the descriptions to **Fig. 4** caption and in **Supplementary**
**Note 5**.

Reviewer #1 (Remarks to the Author):

Wang et al. has addressed most of my questions and concerns. I would like to recommend its publication in Nature Communications. Some minor revisions might be appropriate.

(1) It is better to explicitly state whether the field first modify the lattice then leads to the SHG, or the field directly modify the skyrmions to generate SHG.

(2) By subwavelength structures, I mean the subwavelength confinement, geometric factors and inter-domain wall reflections and (e.g. F-P type) interferences, not the plasmonic aspect.

Reviewer #2 (Remarks to the Author):

The authors have addressed some of my questions well. However, it looks a bit unclear to me what stage they are at to improve the manuscript.

The authors demonstrate that the key advantage with polar skyrmions (and likely other types of topological structures as well) lies in their characteristic nanoscale domain evolution mechanisms. But there is still no solid evidence that show the nanoscale domain evolution in the present version.

As reviewer 1 mentioned, it is fundamental important to establish field-induced transition paths of the skyrmions. It is undoubtable that spectroscopic methods can be applied to reveal motions of matter, but it is hard to say the change in synchrotron X-ray diffraction can be linked with the polarization changes of nanoscale domain evolution of the skyrmions.

I understand that it is challenging to perform the in-situ STEM analysis, as the FIB-prepared samples may suffer breakdown at low voltages (less than 5 V). However, they also mentioned that polar skyrmions can collectively expand or shrink at low to moderate external voltages before switching to a single c-domain at large voltages.....So, maybe the polarization changes will take place before sample breakdown. It possible, I-t mode may be a good choice as presented in Nature, 613, 26, 656 (2023).

Also, it is recommended to show the raw HAADF image and the corresponding polar map of Fig. 1e.

Reviewer #3 (Remarks to the Author):

The authors have responded to all my comments and provided additional information to corroborate their claims.

I recommend to publish the revised manuscript in Nature Communications.

Response to the reviewers' comments

Manuscript ID: NCOMMS-23-49620B

Manuscript Title: Giant electric field-induced second harmonic generation in polar skyrmions

1. Reviewer #1 (Remarks to the Author):

Wang et al. has addressed most of my questions and concerns. I would like to recommend its publication in Nature Communications. Some minor revisions might be appropriate.

Thank you very much for the recommendation and constructive comments.

1.1 Reviewer #1's Comment 1:

It is better to explicitly state whether the field first modify the lattice then leads to the SHG, or the field directly modify the skyrmions to generate SHG.

Authors' Response:

By comparing the skyrmion-hosting superlattice with single layer PbTiO₃ film which exhibits no EFISH modulation, we believe that the applied electric field directly modifies the skyrmions to generate SHG. We have explicitly stated this in the second revised manuscript. Please see **Page 9** (marked in red): "The applied out-of-plane electric field directly couples with the PbTiO₃ lattice and causes switching of its electric dipoles to form larger portion of *c*-domains. Therefore, the skyrmions constituted by those dipoles expand or shrink according to the field direction, displacing their wall positions at which the inversion centers are located (non-zero net P_z)."

1.2 Reviewer #1's Comment 2:

By subwavelength structures, I mean the subwavelength confinement, geometric factors and inter-domain wall reflections and (e.g. F-P type) interferences, not the plasmonic aspect.

Authors' Response:

Thanks for clarifying the question, which we did not fully grasp. We agree that the geometric factors and interferences play important roles in SHG generation of materials

and devices. Subwavelength structures, in our case, is related to the thickness of each
PbTiO_3 or SrTiO_3 layers and the size of skyrmions. Since both of them are far smaller
than the probing wavelength ($\sim 6\text{-}8$ nm vs. 800 nm), we have not learned any approaches
to effectively treat this problem. In fact, we do hope that our work, once published, can
inspire researchers from nonlinear optics communities who might devise new ideas or
theoretical approaches. A deep-nanoscale ordered nonlinear optic system is highly usual,
and polar skyrmion provides such a unique model system (which is the key novelty and
significance of our study).

We have revised the related discussion in **Page 10** as “We further point out that
other extrinsic mechanisms, such as light interference and reflection by the skyrmion
walls^{48,49} and superlattice interfacial effects⁵⁰, might also influence the EFISH response
here, but given the large difference between the skyrmion size and probing wavelength,
an effective approach is currently lacking to substantiate these subwavelength effects.”

**2. Reviewer #2 (Remarks to the Author):**

The authors have addressed some of my questions well. However, it looks a bit
unclear to me what stage they are at to improve the manuscript.

**Authors' Response:**

We greatly appreciate you for offering new constructive comments, and we are
happy to further address them.

The authors demonstrate that the key advantage with polar skyrmions (and likely
other types of topological structures as well) lies in their characteristic nanoscale
domain evolution mechanisms. But there is still no solid evidence that show the
nanoscale domain evolution in the present version.

As reviewer 1 mentioned, it is fundamental important to establish field-induced
transition paths of the skyrmions. It is undoubtable that spectroscopic methods can be
applied to reveal motions of matter, but it is hard to say the change in synchrotron X-
ray diffraction can be linked with the polarization changes of nanoscale domain
evolution of the skyrmions.

**Authors' Response:**

First, we have to clarify that the key novelty and significance of this manuscript is
not the characteristic nanoscale domain evolution mechanisms, but rather discovery of
highly tunable nonlinear optical properties of polar skyrmions and its technological
implications, that is, the tunable SHG accompanied by the skyrmion evolution. The
electric field-induced topological phase transition of polar skyrmions has already been
addressed in a previous study (*Nature Materials* 20, 194, 2021). Their methodology is
a combination of *in-situ* XRD, dielectric measurements and phase-field simulations
(without *in-situ* STEM). Our synchrotron XRD results are consistent with theirs (Figure
4 in the paper) in two key aspects: expansion of the out-of-plane lattice parameter and
reduction in the satellite diffraction intensities, substantiating an electric field-induced
polarization switching to *c*-domains at the expense of polar skyrmions. These authors
state “...when an electric field is applied along the out-of-plane direction
(parallel/antiparallel to the uniform polarization of the skyrmion cores), the skyrmions
progressively expand (shrink) for parallel (antiparallel) fields and ultimately the entire
material becomes uniformly poled with increasing electric field...” Our study builds on
this understanding as well as a number of related studies on topological polar structures.

In fact, by “spectroscopic methods can be applied to reveal motions of matter”, we
previously meant to refer to SHG. SHG is a well-established method to probe structure
of materials and has been applied alone for such purpose in many studies. In our study,
SHG bias spectroscopy and polarimetry yield a wealth of microscopic information
reinforced by synchrotron XRD and phase-field simulations, altogether amounting to
compelling evidence that deepens the understanding of the phase transition behavior of
polar skyrmions. We believe that all these data are self-consistent. Moreover, the
evolution of skyrmions under external field by *in situ* TEM is studied by Zhu *et al.*
(*PRL* 129, 107601, 2022), and our results agrees qualitatively well with their
observations.

We have revised the related discussion in **Page 8** as “Note also that, the field
induced skyrmion evolution path in this study agrees qualitatively well with several
previous studies with indirect measurements and theoretical calculations^{30,43}, as well as
direct *in situ* TEM observations³⁶.”

I understand that it is challenging to perform the in-situ STEM analysis, as the
FIB-prepared samples may suffer breakdown at low voltages (less than 5 V). However,
they also mentioned that polar skyrmions can collectively expand or shrink at low to
moderate external voltages before switching to a single c-domain at large
voltages.....So, maybe the polarization changes will take place before sample
breakdown. It possible, I-t mode may be a good choice as presented in Nature, 613, 26,
656 (2023).

Authors' Response:

**Figure R1** *In-situ* STEM results measured under a bias voltage of 0 V (a,b,c) and 4 V
(d,e,f), showing (a,d) HAADF images, and corresponding (b,e) electric field magnitude
and (c,f) direction images, derived from the measured DPC images. Scale bar = 5 nm.

We now present typical *in-situ* STEM results that we have attained so far in **Fig.**
**R1**, which indeed show signs of polarization changes at 4 V. Because the HAADF
imaging conditions were not so optimized here (as we focused on testing the breakdown
voltage), it is not possible to calculate polar displacement vectors and reconstruct polar
skyrmions from the images. Nevertheless, the DPC images show contrast changes in
the internal electric field distribution (see *e.g.* the boxed region in **Fig. R1c,f**). As the
electric field distribution is correlated with skyrmions, such changes can suggest field-
induced variations of the skyrmions in this region.

Thanks also for appreciating the challenges of *in-situ* electric field STEM.
Compared to single-layer BiFeO₃ as in Nature 613, 26, 656 (2023), the polar skyrmion
system can be even more demanding for its complex polarization configuration. In any
case, we will endeavor to work on it and report the results in a separate article in the
future. We will also try to develop the I-t measurement mode in our setup once we solve
the sample issue.

In addition, we also performed I-V measurement using CAFM (Asylum Research
Cypher, ASYELEC-01-R2 tips) right after your first-round comments came in. As
shown in **Fig. R2**, within the accessible bias range of ± 10 V, no obvious current signals
could be observed above the noise floor. Presumably this is because of the small contact
area by using CAFM tips to collect conduction current and the large film thickness (~ 96
138 nm). For this reason, we had chosen to present the macroscopic I-V results (measured
on 150 μm diameter electrodes) in the first response letter.

**Figure R2** CAFM I-V curve measured directly on the superlattice surface.

Also, it is recommended to show the raw HAADF image and the corresponding
polar map of Fig. 1e.

**Authors' Response:**

Thanks for this suggestion. We have modified Fig. 1e-g (the original Fig. 1g moved
to supplementary materials) and revised the figure caption accordingly. For comparison,
we have also overlaid the polar vector map with the phase-field simulation results.

**3. Reviewer #3 (Remarks to the Author):**

The authors have responded to all my comments and provided additional
information to corroborate their claims.

I recommend to publish the revised manuscript in Nature Communications.

We are deeply grateful for the reviewer's recommendation.

Reviewer #2 (Remarks to the Author):

The authors have addressed my comments well. I recommend to publish the revised manuscript in Nature Communications.